# Reexamining the Kuleshov effect: Behavioral and neural evidence from authentic film experiments

Zhengcao Cao[1,2], Yashu Wang[1]*, Liangyu Wu[1], Yapei Xie[2], Zhichen Shi[1], Yiren Zhong[1], Yiwen Wang[1]*

1 School of Arts and Communication, Beijing Normal University, Beijing, China, 2 State Key Laboratory of Cognitive Neuroscience and Learning, Beijing Normal University, Beijing, China

* viiawang@bnu.edu.cn (YW); wyiw@bnu.edu.cn (YW)

**Data Availability Statement:** The experimental data presented in this article are accessible on Github at the following link: https://github.com/Michaelcao92/KuleshovExp.

## Abstract

Film cognition explores the influence of cinematic elements, such as editing and film color, on viewers' perception. The Kuleshov effect, a famous example of how editing influences viewers' emotional perception, was initially proposed to support montage theory through the Kuleshov experiment. This effect, which has since been recognized as a manifestation of point-of-view (POV) editing practices, posits that the emotional interpretation of neutral facial expressions is influenced by the accompanying emotional scene in a face-scene-face sequence. However, concerns persist regarding the validity of previous studies, often employing inauthentic film materials like static images, leaving the question of its existence in authentic films unanswered. This study addresses these concerns by utilizing authentic films in two experiments. In Experiment 1, multiple film clips were captured under the guidance of a professional film director and seamlessly integrated into authentic film sequences. 59 participants viewed these face-scene-face film sequences and were tasked with rating the valence and emotional intensity of neutral faces. The findings revealed that the accompanying fearful or happy scenes significantly influence the interpretation of emotion on neutral faces, eliciting perceptions of negative or positive emotions from the neutral face. These results affirm the existence of the Kuleshov effect within authentic films. In Experiment 2, 31 participants rated the valence and arousal of neutral faces while undergoing functional magnetic resonance imaging (fMRI). The behavioral results confirm the Kuleshov effect in the MRI scanner, while the neural data identify neural correlates that support its existence at the neural level. These correlates include the cuneus, precuneus, hippocampus, parahippocampal gyrus, post cingulate gyrus, orbitofrontal cortex, fusiform gyrus, and insula. These findings also underscore the contextual framing inherent in the Kuleshov effect. Overall, the study integrates film theory and cognitive neuroscience experiments, providing robust evidence supporting the existence of the Kuleshov effect through both subjective ratings and objective neuroimaging measurements. This research also contributes to a deeper understanding of the impact of film editing on viewers' emotional perception from the contemporary POV editing practices and neurocinematic perspective, advancing the knowledge of film cognition.

**Funding:** This study was financially sponsored through The Arts Project of 2019 National Social Science Fund of China (grant no.19BC041) and The Arts Project of 2023 National Social Science Fund of China (grant no.23ZD07). The Arts Project of 2019 National Social Science Fund of China (grant no.19BC041) had role in study design, data collection and analysis, decision to publish. The Arts Project of 2023 National Social Science Fund of China (grant no.23ZD07) had role in preparation of the manuscript. The authors thank them for their financial support.

**Competing interests:** The authors have declared that no competing interests exist.

## 1. Introduction

Film cognition explores the intersection of cinematic elements and viewer psychology, focusing on how features such as camera angles, editing techniques, and color influence audience perception [1–7]. Through behavioral and neuroimaging experiments [8–10], research in this field examines the impact of these film components on viewers' interpretation of film narratives, contributing to advancing film theory and practice [11].

Reflecting on film history, a notable example of how editing influences viewers' perception is the Kuleshov effect. Identified through an experiment conducted in the 1920s by Soviet filmmaker Lev Kuleshov (1899–1970), this study juxtaposed an image of Russian actor Ivan Mozhukin's neutral face with various emotional contexts in a series of frames arranged in a face-scene-face sequence [12]. This sequence comprises segments featuring an actor with a neutral expression, followed by an emotional scene, and concludes with a repetition of the actor's neutral expression. During this experiment, viewers were tasked with assessing the emotions portrayed by the actor's face. Despite the actor's facial expression remaining consistent, viewers frequently attributed different emotional states to the actor, deeply linked to the accompanying emotional scenes. For example, when the emotional scene depicts a dead girl lying in a cabin, viewers perceive sadness in the neutral face; conversely, when the scene features a little girl playing with a doll, viewers interpret happiness in the neutral face [13].

Since Kuleshov's initial pilot experiment on the Kuleshov effect in the 1920s [12], references to the Kuleshov effect in film textbooks suggest that the effect is well established [14,15]. However, the loss of original film footage, lack of experimental control, and absence of statistical analysis in the initial experiment [12] make it unclear whether the Kuleshov effect truly exists. Several scientific experiments have attempted to replicate the study to address this question, utilizing diverse experimental materials and paradigms throughout the past century [13,16,17]. Despite these efforts, the results have proven controversial regarding the existence of the Kuleshov effect [13,17].

Initially, Prince and Hensley (1992) [17] attempted to replicate the Kuleshov Experiment paradigm. In Prince's study, each sequence started with a fade-in on an actor's neutral face, followed by a cut to an object (soup, coffin, or child), and then a cut back to the actor's neutral face, concluding with a fade-out. Prince tasked the participants whether the actor's neutral face conveyed emotion. However, most participants perceived no emotion from the actors, and Prince's study did not validate the existence of the Kuleshov effect.

In a subsequent study, Ildirar and Ewing (2018) [18] replicated the Kuleshov effect experiment, categorizing participants into two groups: first-time viewers and experienced viewers. Ildirar edited emotional facial clips and emotional contextual clips together to investigate whether viewers constructed spatiotemporal links between the two clips, recognizing that the two clips conveyed a unified narrative. The study revealed that experienced viewers could construct these links, while first-time viewers could not. This finding validated the existence of the Kuleshov effect.

While several studies predominantly explored the Kuleshov effect from a film-centric perspective [17,18], more recent investigations have employed the Kuleshov paradigm to examine the contextual framing impact on facial emotion recognition from a psychological perspective. Barratt et al. (2016) [13] conducted a study employing the Kuleshov paradigm to explore how emotional contexts influence perceptions of facial stimuli. The experiment utilized neutral faces from the Karolinska Directed Emotional Faces (KDEF) picture set and paired them with online static or dynamic videos to create varied emotional stimuli. Participants were tasked with evaluating the valence and arousal of the neutral faces and additionally categorizing the expressions using nine emotional labels. While Barratt's findings confirmed that participants'

judgments were influenced by these contexts, notably affirming the Kuleshov effect, this influence was predominantly evident in specific emotional categories. Particularly, neutral faces paired with sad contexts were perceived as most negative and least aroused, while those paired with contexts of desire were seen as most positive and most aroused, indicating a complex influence of contextual emotion beyond simple valence and arousal.

Building on this, Calbi et al. (2017) [19] sought to enhance the ecological validity of Barratt's findings [13] by using exclusively dynamic images for reverse shots and introducing inter-trial intervals. This study further explored the contextual effects on emotion perception by asking participants to rate both the valence and arousal, and to explicitly categorize the emotion displayed by neutral faces following a Kuleshov sequence. While the results demonstrated significant effects on both valence and arousal, these effects were most pronounced in fear-inducing contexts. Moreover, participants tended to categorize the neutral facial expressions in a manner congruent with the preceding emotional context. These findings suggest that the emotional context significantly influenced the emotional judgment of neutral faces, resulting in a context-dependent bias of emotion perception.

In a similar vein, Mullennix et al. (2019) [20] rigorously manipulated the Kuleshov effect paradigm, utilizing static facial stimuli and varying presentation times to investigate the effect of visual context on the interpretation of facial expressions. After watching an emotional context, participants viewed a neutral face, and the results indicated that more faces were labeled as "happy" after viewing a pleasant context, and more faces were labeled "sad" or "fearful" after viewing an unpleasant context. In summary, these studies collectively reveal that the assessment of neutral face's emotional expressions is influenced by the preceding emotional context when using static faces and online emotional videos. This suggests that the Kuleshov effect can be detected with conventional psychological experiment materials.

However, when reexamining the Kuleshov effect, the significant disparity between static images and dynamic films, following the definition of film employing moving images to create continuous life experiences [21], cannot be disregarded. Specifically, although the use of zoom-ins with static neutral faces from the KDEF picture set approximates reality [19], it fails to capture the nuanced facial movements inherent in actual film scenes [22,23], resulting in the omission of spatiotemporal information from dynamic faces [24]. Moreover, in authentic films, the continuity of backgrounds between scenes establishes a cohesive spatial context that enables viewers to perceive these scenes as unfolding within the same spatial realm [7,25]. The replication studies predominantly rely on isolated facial expressions and emotional scenes [18–20], overlooking the alignment between facial backgrounds and emotional scenes present in real-world film scenarios. This deviation from authentic film scenarios may compromise the robustness of validating the Kuleshov effect. Therefore, the need to reexamine the Kuleshov effect using high-ecological film materials is compelling. In the current study, we opted for a neutral expression in high-ecological film materials, which rarely appear in commercial fiction films. This choice was made because it aligns with the original Kuleshov experiment, and the performance of a neutral expression can be more easily controlled.

In addition to reexamining the Kuleshov effect through behavioral experiments, the use of cutting-edge neuroimaging techniques can enhance its validation by revealing its neural correlates. There are two distinct approaches to neurocinematic research. The first utilizes traditional experimental methods, notably the General Linear Model (GLM) analysis, based on event- or block-designs. In these designs, parameters for BOLD signal fluctuations indicative of brain activation are predefined [26]. The second approach employs naturalistic stimuli within functional magnetic resonance imaging (fMRI) environments, incorporating complex mise-en-scène and utilizing model-free analytical methods such as Inter-Subject Correlation (ISC). These techniques are essential for identifying brain activations linked to specific,

annotated elements of the stimulus. For instance, Hasson utilized the ISC method and found that Sergio Leone's *The Good, the Bad, and the Ugly* (1966) evoked similar responses across approximately 45 percent of the cortex in viewers during the movie [27]. When reexamining the Kuleshov effect with fMRI, this conventional approach is often more suitable due to the simplicity and clarity of the face-scene-face sequence, making it ideal for event-based designs.

To our knowledge, only two studies have attempted to explore the neural correlates of the Kuleshov effect. To our knowledge, only two studies have attempted to explore the neural correlates of the Kuleshov effect. Mobbs et al. (2006) [10] conducted an fMRI experiment that unveiled specific mechanisms involved in the Kuleshov effect, particularly regarding how emotional scenes impact neutral facial expressions. The study juxtaposed various emotional scenes with neutral facial expressions, revealing activations in distinct brain regions including the bilateral temporal pole (TP), anterior cingulate cortex (ACC), amygdala, and bilateral superior temporal sulcus (STS). The TP, which connects to structures important for processing emotional and social information such as the STS and amygdala, contributes to contextual framing by interpreting facial expressions based on prior emotional contexts. The STS plays a specific role in perceiving the dynamic aspects of faces, while the amygdala is crucial for processing socially relevant information. Additionally, the ACC is involved in schema-based appraisals, further emphasizing its role in emotional regulation. This study provided significant insights into the neural correlates of the Kuleshov effect.

Similarly, Calbi et al. (2019) [28] conducted an electroencephalographic–EEG experiment that revealed specific mechanisms involved in the Kuleshov effect. In the experiment, participants were tasked with rating the valence and arousal of neutral faces while simultaneously undergoing EEG recording, and were subsequently asked to categorize the target person's emotional state. The behavioral results largely replicated those of Calbi et al. (2017) [19]. However, in the Calbi et al. (2019) [28] study, the valence ratings of neutral faces in the happy context were found to be significantly higher than those in the neutral context, augmenting the finding of the contextual influence on the perception of neutral faces, an aspect not observed in the earlier study. EEG results discovered that viewers rated faces as more arousing when preceded by a fearful context. The study pinpointed a high-amplitude late positive potential (LPP) in the EEG data when faces were preceded by emotional contexts, indicating a cognitive process of expectation attribution that influences how facial expressions are interpreted. This was evidenced by the activation of LPP when evaluating incongruent sequences of stimuli. However, these studies have not addressed the neural correlates of the Kuleshov effect within a face-scene-face sequence using authentic films. This gap in research makes it challenging to fully understand how the Kuleshov effect is generated during film watching.

In film history, there are two types of film editing: montage editing and continuity editing. The Kuleshov sequence encompasses both. The Kuleshov sequence is often interpreted as an example of classic montage editing, where the goal is to combine images to generate new meanings—for instance, combining a clip of Mozhukin's neutral face with a clip of a bowl of soup may create the idea of hunger—the original Kuleshov experiment also illuminates another fundamental aspect of film editing: continuity editing. The Kuleshov sequences serve as early examples of continuity editing [13,29], which includes a shot-reverse-shot structure, aiming to maintain a coherent narrative space and time. Moreover, these sequences exemplify point-of-view (POV) editing practices. In the 'glance shot', a character is shown gazing toward an offscreen entity (in this instance, Mozhukin's neutral face), whereas the 'object shot' captures the item being looked at (such as a dead girl in a cabin). The POV structure is easily comprehended by viewers because it mimics the natural human and primate tendency to follow the gaze of an intentional agent [30]. Thus, reexamining the Kuleshov effect, particularly how it employs the POV structure and generates new meanings from neutral faces, contributes not

only to validating classic montage theory but also to revealing the efficacy of contemporary film editing practices. Although the Kuleshov sequence supports classic montage editing, it should be noted that the Kuleshov sequence differs from the montage sequence. The latter incorporates informative titles, newsreel footage, and music to create a quick and regular rhythm that depicts the passage of time [29].

In this study, our main objective is to investigate the existence of the Kuleshov effect using authentic films within the realm of film cognition. In Experiment 1, we validated the existence of the Kuleshov effect with authentic film sequences, employing film cameras to capture actor's neutral facial expressions and emotional scenes. These elements were combined into a face-scene-face sequence, ensuring consistency in the background between faces and emotional scenes in authentic films. Participants were then tasked with rating the emotions portrayed by the actors. Our hypothesis posited that the emotional scene (fearful, happy) would influence the emotional judgment of neutral faces, as evidenced by the attribution of negative or positive valence to these faces. In Experiment 2, we sought to explore the neural correlates of the Kuleshov effect to validate its existence further. Employing identical authentic film sequences, we refined the Kuleshov sequence by incorporating jitters into the procedure to accommodate fMRI data acquisition. Initially, our goal was to observe how viewers perceive new meaning within the Kuleshov effect by contrasting activation between faces. Subsequently, we examined the context-dependent bias of the Kuleshov effect, investigating how brain activity elicited by faces was altered by emotional scenes. Thus, reexamining the Kuleshov effect in this study not only provides empirical support for classic montage theory but also offers insights into the adaptability and impact of contemporary POV editing techniques on emotional perception.

## 2. Methods

The current study adopted the previous Kuleshov paradigm [19], which includes a neutral face, an emotional scene, and a neutral face presented in a face-scene-face sequence. The primary objective was to observe the context-dependent bias in judging the emotions of neutral faces affected by accompanying emotional scenes (fearful, neutral, happy) within authentic films. In Experiment 1, we captured authentic face and scene materials and combined them in film sequences. Subsequently, these film sequences were employed in a behavioral experiment to explore the existence of the Kuleshov effect under laptop experiment conditions. In Experiment 2, the same film sequences were utilized in an fMRI study to revalidate the Kuleshov effect within an MRI scanner environment and to investigate the neural correlates of the Kuleshov effect. Both experiments employed a within-subjects design. According to Cognitive Appraisal Theory [31], a dissociation exists between emotional experiences and physiological reactions. To explore this dissociation, we employed the emotional intensity scale in Experiment 1 to assess the subjective perceived emotional degree in the actor's performance, and the arousal scale in Experiment 2 to measure physiological responses to the neutral face.

We determined the sample size for the current study using G*Power software. We set the effect size $f$ to 0.6, $\alpha$ to 0.05, power (1-$\beta$) to 0.8, and the number of groups to three, representing fearful, neutral, and happy conditions. The output for the total sample size was 30. We ensured that the sample sizes in the behavioral experiment not only exceeded 30 but also surpassed those used in previous studies; specifically, 28 participants by Calbi et al. (2017) [19] and 44 participants by Mullennix et al. (2019) [20]. Similarly, for the fMRI experiment, we ensured that our sample size was greater than 30 and exceeded the 14 participants used by Mobbs et al. (2006) [10].

We recruited a cohort of 102 healthy individuals (51 females, aged 23.01 ± 3.21 years) with normal or corrected-to-normal vision from Beijing Normal University. Eligible participants,

confirmed through a questionnaire, reported no history of panic disorder. Individuals majoring in film studies were deliberately excluded from participation. All provided written informed consent, approved by the ethics committee of the State Key Laboratory of Cognitive Neuroscience and Learning at Beijing Normal University (approval no. IRB B 0030 2019001). Participants were recruited between October 10, 2019, and September 16, 2023. They received monetary compensation, and the aforementioned committee ethically endorsed the experimental procedures.

We allocated 12 participants (6 females, aged 26.75 ± 3.41 years) to the materials rating experiment, 59 participants (29 females, aged 22.81 ± 2.75 years) to Experiment 1 (behavioral experiment), and 31 participants (16 females, aged 21.94 ± 2.99 years) to Experiment 2 (fMRI experiment). It was ensured that there was no participant overlap between the two experimental cohorts, thus maintaining the independence of behavioral and neural data.

The experimental materials presented in this article are accessible in the Supplementary Videos (S1–S9 Videos) and the experimental data can be accessed on GitHub at the following link: https://github.com/Michaelcao92/KuleshovExp.

## 2.1. Experiment 1

**2.1.1. Stimuli.**   To create authentic film sequences for the Kuleshov effect, we carefully designed the camera positions to adhere to the face-scene-face sequence utilizing the shot-reverse-shot structure (S1 Fig). Following the established Kuleshov experiment paradigm, which includes a neutral face, an emotional scene, and a repeated neutral face, we captured one set of face clips and one set of emotional scene clips. All clips were then converted to grayscale, and the sound was removed to eliminate the potential impact of color and sound, in line with prior studies [13,19]. Subsequently, we combined these elements in a face-scene-face film sequence using a shot-reverse-shot structure. This setup presents the actor's face in the "shot" and the corresponding emotional scene in the "reverse shot," illustrating what the actor is supposedly observing. Thus, from a POV editing perspective, the "glance shot" depicts the actor's face, while the "object shot" displays the corresponding emotional scene. Actors involved in this study received approval from the Institutional Review Board of the State Key Laboratory of Cognitive Neuroscience and Learning at Beijing Normal University (approval no. IRB B 0030 2019001). Written informed consent was obtained from the actors before their participation. Additionally, all actors consented to the publication of any potentially identifiable images or data included in this article.

*Neutral Faces*

For the face clips, we employed a cinecamera, lighting equipment, and a blue screen for the film shoot (S1 Fig). The actors were seated before the blue screen and instructed to maintain a neutral facial expression while looking at a focal point adjacent to the camera. Throughout the recording, actors kept their facial muscles relaxed, minimizing movements to blinking only, and we recorded 51 face clips from various actors, each cut into a 2-second clip. Subsequently, we carefully selected 15 male and 15 female actors based on their neutral expressions and ensured similarity in appearance. This selection was achieved through an internal group rating conducted by the authors. The 30 neutral faces were randomly allocated into three groups (fearful, neutral, and happy) for the facial material rating experiment. This experiment aimed to assess the emotional expression of these neutral faces, with detailed information provided in S1 Text. Briefly, twelve participants rated the valence of the actor's neutral faces. The average valence was approximately zero (-0.02 ± 0.14), and no significant differences were observed among the three groups ($F_{2, 8} = 0.40$, $p = 0.682$, $\eta2$ p = 0.091).

*Emotional scenes*

We collaborated with a professional film director and a dedicated crew to capture emotional scenes. Initially, we carefully designed these scenes before shooting, ensuring their

compatibility with the shot-reverse-shot structure while maintaining a consistent 180-degree axis orientation (S1 Fig). Three distinct categories of emotional scenes were conceptualized, each encompassing ten unique contexts (S2 Fig). These scenarios spanned various genres, including horror, documentary, and comedy. Fearful scenes depicted chilling scenarios such as peeping, murder, and a ghost, and. Neutral scenes featured commonplace settings like a printer, a bus stop, and a traffic light. Happy scenes showcased joyful images like food, an attractive girl, and humorous expressions. The balance between biological and non-biological elements varies across conditions: fearful (5:5), neutral (4:6), and happy (6:4). The overall ratio between biological and non-biological elements is approximately 1, ensuring a balanced emotional stimulus. The production quality for each emotional scene was meticulously maintained at a cinematic level, involving coordination with the film director, lighting setup, acting, and cinematography. After capturing the raw footage, we implemented color correction to impart a cinematic feel to the videos. The videos were then edited into 4-second clips to serve as emotional scene materials in the Kuleshov experiment. The clip length aligns with the Average Shot Length (ASL) found in mainstream Hollywood films, typically between 3 and 4 seconds [19,32,33]. To assess the emotional impact of these scenes, a scene rating experiment was conducted involving the same twelve participants who took part in the facial rating experiment (S1 Text). The results indicated negative valence for fearful scenes (-2.19 ± 0.24), positive valence for happy scenes (1.98 ± 0.17), and neutral valence for neutral scenes (0.48 ± 0.08), with significant differences observed among the three groups ($F_{2,\ 8}$ = 107.92, $p < 0.001$, $\eta2$ p = 0.964). *Post hoc* tests using Bonferroni correction confirmed significant differences in valence among each group ($p < 0.001$).

*Final stimuli*

For each trial in the Kuleshov experiment, we selected a clip from the neutral facial materials and a clip from the emotional scenes. We combined them in the sequence of face-scene-face, with the identical face appearing twice in the sequence. Additionally, we ensured that each facial clip and emotional scene clip was used only once. This resulted in a total of 30 combined film sequences (Fig 1A). To maintain a gender balance of neutral faces, each emotional scene included five male and five female actors. Following prevalent conventions found in mainstream films, we paired genders between neutral faces and emotional scenes. To create a more authentic film scenario, ensuring that the backgrounds among different video clips were similar was crucial. This background matching enhanced the ecological design of the Kuleshov validation experiment and addressed a previously overlooked aspect in similar studies [18,19]. In the matching process, blue screen image matting was utilized to extract the actors. Subsequently, the backgrounds were substituted with surrounding videos or images, all of which were shot along with emotional scenes (S3 Fig). Notably, most videos and images were captured simultaneously with the emotional scenes. Adjustments were made to the color temperature, brightness, and contrast of the facial videos to harmonize them with the emotional scenes. Finally, we converted the color facial video into black and white [19]. A visual comparison between highly authentic film scenarios using matched backgrounds and less authentic film scenarios using unmatched backgrounds is shown in (S4 Fig), indicating that the current film sequence provides a more authentic and immersive experience than the unmatched background sequence. The final film sequences are demonstrated in Supplementary Videos (S1–S9 Videos). The final film sequences were presented in MP4 format and the resolution of the film sequence was 1920 pixels × 1080 pixels.

**2.1.2. Subjective scales.** The experiment employed the valence scale and emotional intensity scale to assess emotional perception. Valence refers to the positive or negative quality of an emotion as depicted in the actor's performance, while emotional intensity measures the perceived degree of emotion in the actor's performance. Both scales were Visual Analogue Scales,

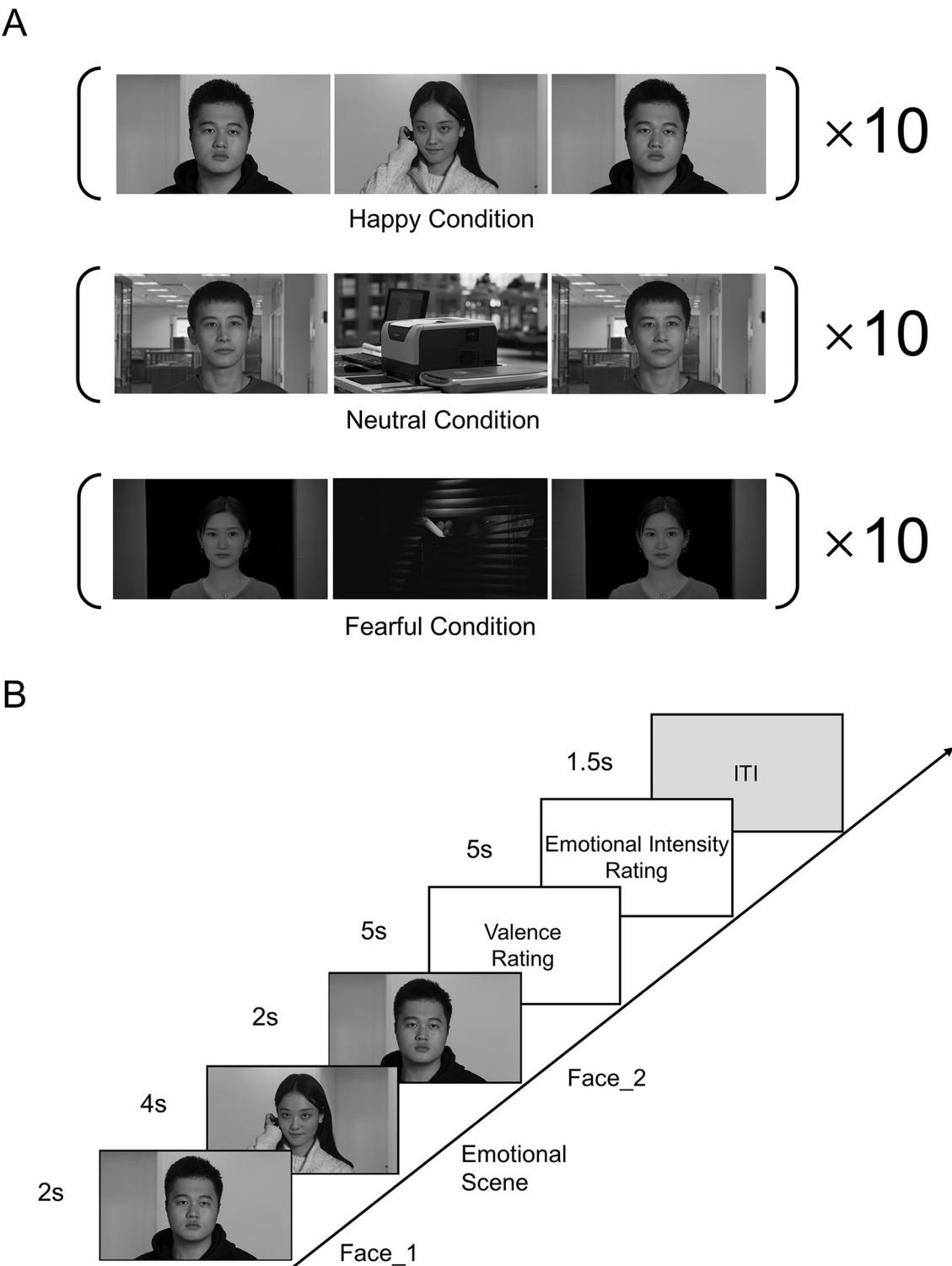

**Fig 1. Experimental materials and procedure.** (A) Examples of combined film sequences in happy, neutral, and fearful conditions. Each condition comprises ten trials. (B) The procedure for Experiment 1 is illustrated. Each trial started with a 2-second clip featuring a neutral face, followed by a 4-second clip depicting an emotional scene, and concluded with another 2-second clip of the same neutral face. After viewing the film sequence, participants were instructed to rate the valence and emotional intensity of the neutral face portrayed by the actor. The trial concluded with a 1.5-second inter-trial interval (ITI).

adapted from the Self-Assessment Manikin (SAM) [34]. To prepare the valence scale, the most negative, neutral, and positive expressions from the first line of the original SAM were selected and labeled from -1 to 1 beneath these pictures. Similarly, for the emotional intensity scale, the images from the second line of the original SAM were chosen, their order reversed, and they were labeled from 1 to 5 beneath these pictures.

**2.1.3. Procedure.** Before the experiment, 59 participants were briefed on the procedure of the experimental task and the meanings of valence and emotional intensity. They then completed two practice trials to familiarize themselves with the procedure, using film sequences different from those in the formal experiment. If participants had questions about the procedure or the meanings of the scales, the instructions were repeated. In each trial, participants initiated the session by viewing a 2-second clip featuring a neutral face, succeeded by a 4-second clip of an emotional scene, and concluded with another 2-second clip of the same neutral face (Fig 1B). Following the 8-second film presentation, participants were instructed to rate the neutral face portrayed by the actor using the two distinct scales. The valence scale required participants to rate the valence of the neutral face on a scale from -1 to 1, where -1 denoted a fearful emotion, 0 represented a neutral emotion, and 1 indicated a happy emotion. The emotional intensity scale required participants to rate the emotional intensity of the neutral face from 1 to 5, with 1 signifying low emotional intensity and 5 representing high emotional intensity. Following the 8-second film presentation, participants were instructed to rate the neutral face portrayed by the actor using two distinct scales. The first scale required participants to rate the valence of the neutral face on a scale from -1 to 1, where -1 denoted a fearful emotion, 0 represented a neutral emotion, and 1 indicated a happy emotion. On the second scale, participants rated the emotional intensity of the neutral face from 1 to 5, with 1 signifying low emotional intensity and 5 representing high emotional intensity. Participants were instructed to rate by clicking the corresponding scale value with a mouse. Each rating had a 5-second response time. Subsequently, a 1.5-second inter-trial interval (ITI) concluded the trial, and participants seamlessly progressed to the subsequent trial, completing 30 trials (Fig 1A).

Stimuli delivery and response recording were controlled using PsychoPy 3.2 software (https://www.psychopy.org/), administered on a 14-inch laptop. To minimize potential influences on the Kuleshov effect, participants were queried with two questions upon completing the experiment: (1) "Have you heard of the Kuleshov effect?" and (2) "How many movies do you watch per week on average?" Additionally, to maintain balanced emotional conditions, half of the participants followed the order 'fearful–neutral–happy,' while the other half followed 'happy–neutral–fearful.' Gender distribution was also balanced among the subgroups.

**2.1.4. Data analysis.** We hypothesized that the existence of the Kuleshov effect would be evident if emotional scenes influence the rating of emotions on neutral faces, leading to a valence score different from 0. More precisely, we anticipate that fearful scenes will prompt a negative emotional rating for neutral faces (valence score tending to -1). In contrast, happy scenes evoke a positive emotion (valence score tending to 1). To test this, we computed the average valence for all participants under each type of emotional condition, serving as an indicator of the degree of the Kuleshov effect. Subsequently, a repeated-measures analysis of variance (ANOVA) was conducted to identify statistical differences in valence among the three emotional conditions. *Post hoc* comparisons among the means were performed, employing a Bonferroni correction to account for multiple comparisons. The analysis was performed using IBM SPSS Statistics 24 software (https://www.ibm.com/spss?lnk=flatitem), which was similarly employed for analyzing emotional intensity. As for the subgroups with knowledge of the Kuleshov effect, an Independent Samples Mann-Whitney U Test was conducted due to the unmatched sample sizes for each subgroup. A similar Independent Samples Mann-Whitney U Test was conducted for subgroups based on film-watching frequency.

## 2.2 Experiment 2

2.2.1. Subjective scales

The experiment employed the valence scale and arousal scale to assess emotional perception. Valence refers to the positive or negative quality of an emotion as depicted in the actor's performance, while arousal measures the viewer's physiological response to the actor's neutral face. Both scales were VAS, adapted from the SAM [34]. To prepare the valence scale, the images from the first line of the original SAM were chosen, their order was reversed, and they were labeled from -4 to 4 beneath these pictures. Similarly, for the arousal scale, the images from the second line of the original SAM were also chosen, their order was reversed, and they were labeled from 1 to 9 beneath these pictures.

**2.2.2. Procedure.** In this experiment, authentic film sequences from Experiment 1 were employed. To distinguish blood-oxygen-level-dependent (BOLD) activity between neutral faces and emotional scenes [10], the Kuleshov sequence was strategically modified. Randomized jitters, characterized by the introduction of variable timing intervals, were incorporated between neutral faces and emotional scenes (Fig 2). This modification aims to enhance the separation of BOLD responses associated with different stimuli, providing a more accurate representation of neural activity behind the Kuleshov effect.

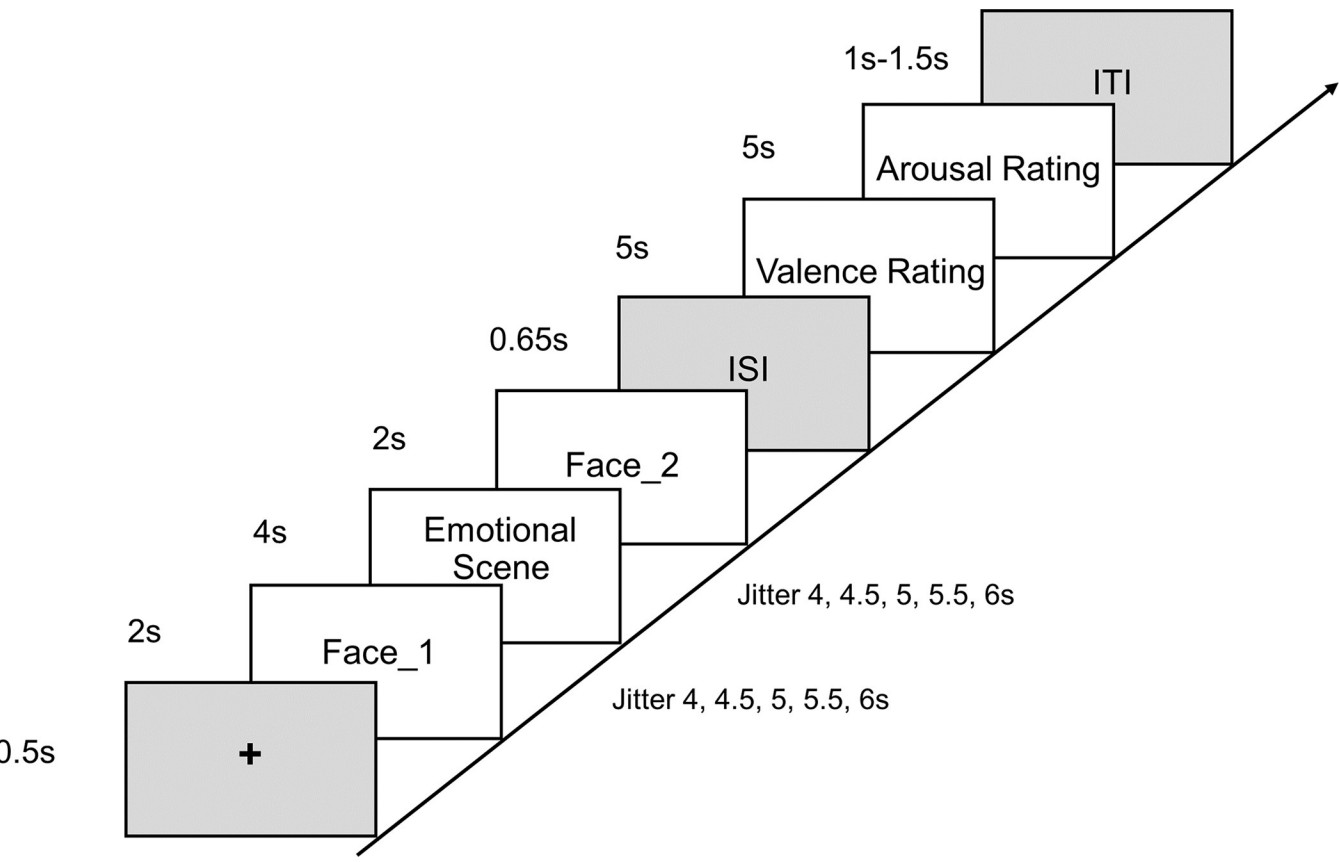

**Fig 2. Experimental procedure in experiment 2.** Each trial was initiated with a 0.5-second presentation of a crosshair sign, followed by a 2-second presentation of a neutral face. This was followed by a variable jitter lasting 4 to 6 seconds, which in turn was followed by a 4-second presentation of an emotional scene. Another jitter lasting 4 to 6 seconds followed, and then a 2-second presentation of the similar neutral face. Following the film sequence presentation, a 0.65-second inter-stimuli interval (ISI) ensued, after which participants were required to rate the valence and arousal of the neutral face portrayed by an actor. Each rating had a 5-second response time. The trial concluded with a 1 to 1.5-second ITI.

Before the experiment, 31 participants were briefed on the procedure of the experimental task and the meanings of valence and arousal. They then completed two practice trials to familiarize themselves with the procedure, using film sequences different from those in the formal experiment. If participants had questions about the procedure or the meanings of the scales, the instructions were repeated. Subsequently, participants underwent T1-weighted image scans and then completed the Kuleshov sequence during fMRI scans. In each trial, participants viewed 10 seconds of experiment instructions before the first trial. Each trial was initiated with a 0.5-second crosshair sign, followed by 2 seconds of neutral face presentation. A random jitter lasted 4 to 6 seconds, followed by 4 seconds of emotional scene presentation, another 4 to 6 seconds of jitter, and then 2 seconds of similar neutral face presentation. Following the film sequence presentation, there was a 0.65-second inter-stimuli interval (ISI). Participants then rated the valence and arousal of the neutral face portrayed by actor. Each rating had a 5-second response time. The valence scale ranged from -4 (negativity) to 4 (positivity), and the arousal scale ranged from 1 (low arousal) to 9 (high arousal). The purpose of the expanded scale range in Experiment 2 was to enhance the sensitivity of our measurements and improve the robustness of our statistical analyses by reducing the potential for ceiling or floor effects. Participants were instructed to rate by clicking the corresponding scale value with fMRI-compatible keyboards. Participants first used the two keys on the left-hand keyboard to select the value, then used the single key on the right-hand keyboard to confirm their selection. Subsequently, an ITI lasting 1 to 1.5 seconds concluded the trial. After completing one trial, participants proceeded to the next, resulting in a total of 30 trials.

Stimuli delivery and response recording were controlled using PsychoPy 3.2 software. To minimize potential influences on the Kuleshov effect, participants were queried after the experiment: "Have you heard of the Kuleshov effect?" Furthermore, we ensured a balanced order of emotional conditions. As there were three emotional conditions, each condition included ten trials. The ten trials for each emotional condition were presented together, while the order of emotional conditions was randomized across participants. Each participant experienced one of the six sequences: 'fearful-neutral-happy', 'fearful-happy-neutral', 'happy-neutral-fearful', 'happy-fearful-neutral', 'neutral-fearful-happy', or 'neutral-happy-fearful'. Additionally, the trials within each emotional condition were also randomized. To mitigate the risk of emotional carryover effects from one trial to the next, we included a jittered ITI between each trial, allowing sufficient time for participants' emotional states to return to baseline. This approach helps ensure that the emotions induced by one trial do not influence subsequent trials.

**2.2.3. Data acquisition.**   In this experiment, we used a Siemens 3T Prisma MRI scanner for scanning. Magnetization Prepared Rapid Acquisition Gradient-echo (MPRAGE) imaging was employed to scan the 31 participants and acquire their three-dimensional T1-weighted data. The imaging parameters were as follows: repetition time (TR) / echo time (TE) / inversion time (TI) = 2530 ms / 2.27 ms / 1100 ms; flip angle (FA) = 7˚; field of view (FOV) = $256 \times 256$ mm$^2$; number of slices = 208; slice thickness = 1 mm; voxel size = $1 \times 1 \times 1$ mm$^3$; The acquisition time for structural image data was 8 minutes. Task-related fMRI scans were performed using a T2-weighted echo planar imaging sequence (EPI/separate). The imaging parameters were as follows: TR / TE = 2000 ms / 34 ms; FA = 70˚; FOV = $200 \times 200$ mm$^2$; matrix size = $100 \times 100$; number of slices = 72; slice thickness = 2 mm; slice gap = 0 mm; voxel size = $2 \times 2 \times 2$ mm$^3$; volume number = 480. Fild map parameters were as follows: TR / TE1 / TE2 = 720 ms / 4.92 ms / 7.38 ms; FA = 70˚; FOV = $200 \times 200$ mm$^2$; matrix size = $100 \times 100$; number of slices = 72; slice thickness = 2 mm; slice gap = 0 mm; voxel size = $2 \times 2 \times 2$ mm$^3$; volume number = 1.

**2.2.4. Data analysis.**

*Behavioral data*

In the analysis of behavioral data, the primary goal was to examine the existence of the Kuleshov effect in the MRI scanner. The analytical approach for behavioral data mirrored that

of Experiment 1. Average valence and arousal scores were calculated for all participants across each type of emotional condition. Statistical differences among the three emotional conditions were assessed using repeated-measures analysis of variance (ANOVA), and *post hoc* comparisons among the means were conducted, employing a Bonferroni correction to account for multiple comparisons.

### Preprocessing of fMRI data

In this experiment, we preprocessed the brain imaging data from the 31 participants using SPM12 software (https://www.nitrc.org/projects/spm/). The preprocessing steps included: 1) Converting scanned DICOM format data to NIFTI format. 2) Time-slice correction to rectify time differences between different slices. 3) Field map correction to address geometric distortions caused by magnetic field inhomogeneity. Two different echo times (TE) of 4.92 milliseconds and 7.38 milliseconds were used, and distortion correction was performed using phase and magnitude images. 4) Head motion correction and distortion correction, with a motion correction setting of a quality parameter at 0.9 and a 4 mm separation, Gaussian kernel parameters (FWHM 5 mm × 5 mm × 5 mm). 5) Registration of structural and functional images, utilizing normalized mutual information (NMI) as the cost function and setting spatial separation parameters and Gaussian kernel parameters (FWHM 7 mm × 7 mm × 7 mm). 6) Segmentation of T1 images into different tissue types such as scalp, skull, cerebrospinal fluid, gray matter, and white matter. 7) Spatial normalization, aligning functional images to the MNI standard template and resampling to 2 mm × 2 mm × 2 mm. 8) Spatial smoothing with Gaussian kernel parameters (FWHM 4 mm × 4 mm × 4 mm) to enhance signal-to-noise ratio for group-level analysis.

### Analysis of fMRI data

In the fMRI data analysis, our investigation into the neural correlates of the Kuleshov effect comprised two primary objectives. Firstly, we aimed to identify neural correlates associated with creating new meaning on the neutral face within the Kuleshov sequence. To achieve this, we contrasted the neural activity evoked by Face_2 and Face_1 in each condition, assuming the existence of the Kuleshov effect. Secondly, we sought to uncover the neural correlates of the Kuleshov effect bias, which leads to a context-dependent bias in emotional perception. This was achieved by comparing the neural activity elicited by Face_2 in the fearful or happy condition with that in the neutral condition. We then examined the interactions between neutral face processing and emotional conditions by comparing the (Face_2 – Face_1) contrast in emotional conditions with the corresponding contrast in the neutral condition. This allowed us to detect unique brain regions contributing to negative or positive emotional perception bias, providing further evidence for the existence of the Kuleshov effect.

To accomplish these objectives, we conducted first-level SPM analyses on each participant using a general linear model to obtain contrast activation. The first-level SPM analysis involved averaging activation differences across ten trials of the same emotional condition. Subsequently, these contrast images were entered into second-level SPM analyses (one-sample T-tests) to evaluate the main effect of each contrast. The results of the second-level analysis were obtained in the form of spm_T maps. Subsequently, further statistical analysis was conducted using xjView software (https://www.alivelearn.net/xjview/). This involved selecting positively activated brain regions and establishing the default cluster size threshold of five in xjView, which is deemed appropriate for our study to ensure that no critical activated regions are overlooked, necessary for understanding the neural mechanisms underlying the Kuleshov effect. Additionally, we applied false discovery rate (FDR) correction for multiple comparisons and set the significance level at $p < 0.05$. Following these steps, we reported the locations of brain regions with FDR multiple comparison correction.

## 3. Results

### 3.1. Experiment 1

In Experiment 1, our primary goal was to ascertain the existence of the Kuleshov effect under the laptop experiment condition, which manifests as the perception of emotions from neutral faces influenced by the accompanying emotional scenes. During the material rating experiment, neutral faces received an average valence rating close to zero (0.02 ± 0.13), indicating a neutral baseline.

Upon incorporating these neutral faces into the Kuleshov sequence, the emotional perception of these neutral faces was significantly affected by the accompanying emotional scene. Specifically, in the fearful condition, viewing fearful scenes led to a negative valence rating for neutral faces (-0.45 ± 0.04), indicating a perceived negative emotion. Conversely, in the happy condition, where scenes evoked positive emotions, a positive valence rating for neutral faces was observed (0.27 ± 0.04). Importantly, in the emotionless condition, neutral scenes did not induce any emotional perception, reflected in a neutral valence rating for neutral faces (0.07 ± 0.03), as illustrated in Fig 3A. The results affirm that viewers attribute emotions to neutral faces.

To assess whether the Kuleshov effect introduced a context-dependent bias in the emotional perception of neutral faces, we conducted a repeated-measures ANOVA on the three emotional conditions (fearful, neutral, happy). Mauchly's Test of Sphericity indicated a violation of the sphericity assumption ($p < 0.001$), prompting the use of the Greenhouse-Geisser correction, which revealed a significant main effect of emotional conditions on valence ($F_{1.6, 92.5} = 104.05$, $p < 0.001$, $\eta2$ p = 0.642). Subsequent *post hoc* tests using Bonferroni correction confirmed significant differences in valence among each condition ($p < 0.001$), supporting the existence of the Kuleshov effect.

Despite emotional scenes influencing valence ratings, no corresponding bias was observed in emotional intensity ratings. A repeated-measures ANOVA on emotional intensity showed that Mauchly's Test of Sphericity was not significant ($p = 0.071$), indicating that the assumption of sphericity was met. The main effect of emotional conditions on emotional intensity was not significant ($F_{2, 57} = 0.13$, $p = 0.879$, $\eta2$ p = 0.005, shown in Fig 3B), suggesting that participants perceived similar emotional intensity in the actor's performances across different emotional conditions.

Lastly, out of the 59 participants, seven already knew about the Kuleshov effect, while 52 did not. Additionally, 42 participants watched 0–2 films per week, 16 watched 3–5 films per week, and only one watched 6–8 films per week. Prior knowledge of the Kuleshov effect or viewing frequency did not impact the observation of the Kuleshov effect (S5 and S6 Figs).

### 3.2. Experiment 2

**3.2.1. Behavioral data.** In this analysis, we sought to revalidate the existence of the Kuleshov effect within an MRI scanner environment. Fearful scenes elicited a negative valence rating on neutral faces (-1.48 ± 0.17), while happy scenes generated a positive valence rating on neutral faces (0.89 ± 0.16). However, neutral scenes did not induce an emotional bias in valence ratings on neutral faces (-0.16 ± 0.07), as illustrated in Fig 3C. Subsequently, a repeated-measures ANOVA was conducted. Mauchly's Test of Sphericity indicated a violation of the sphericity assumption ($p = 0.005$), prompting the use of the Greenhouse-Geisser correction. This correction revealed a significant main effect of emotional conditions on valence ($F_{1.5, 45.9} = 60.10$, $p < 0.001$, $\eta2$ p = 0.667), with *post hoc* tests using Bonferroni correction revealing significant differences in valence among each condition ($p < 0.001$). These results reaffirmed the existence of the Kuleshov effect in an MRI scanner.

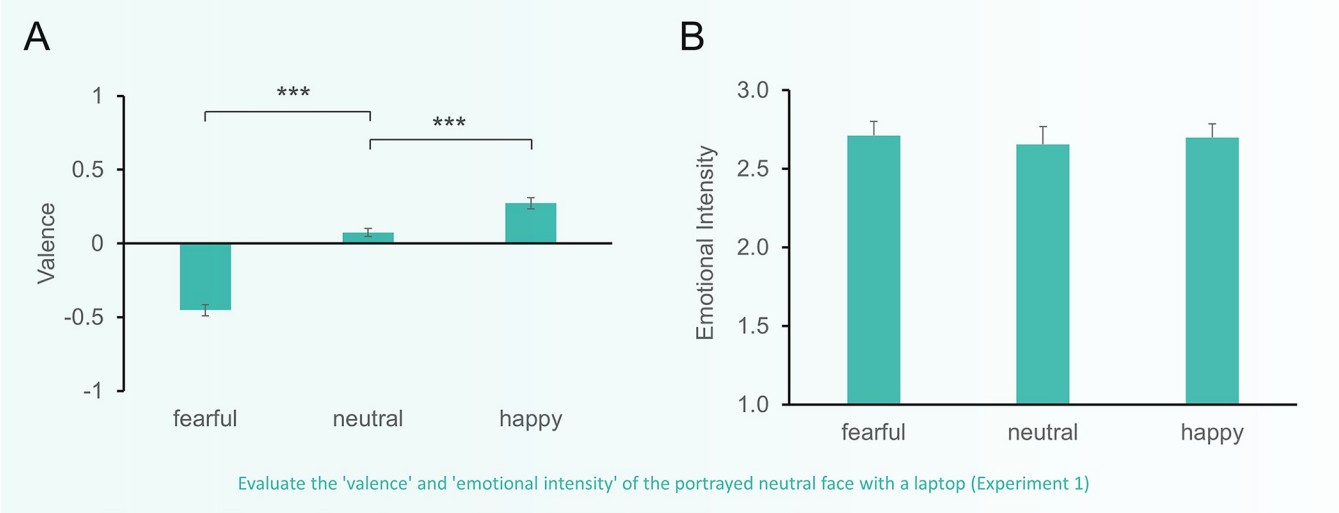

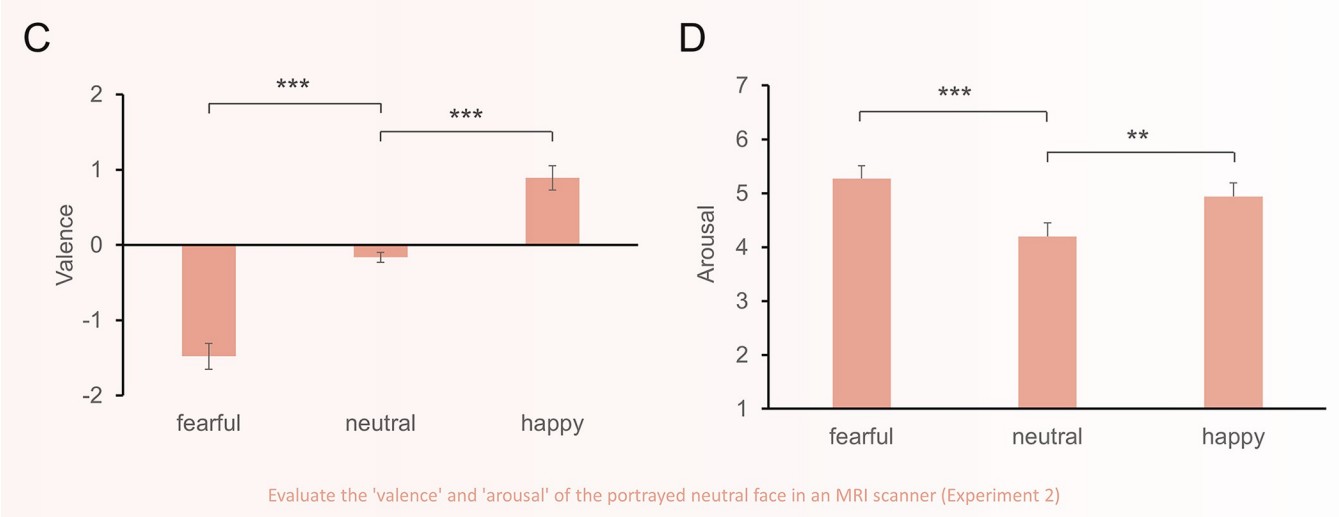

**Fig 3. Behavioral results.** (A) Averaged valence scores from 59 participants in Experiment 1. A significant main effect of emotional conditions is observed ($F_{1.6, 92.5} = 104.05$, $p < 0.001$, $\eta 2$ p = 0.642, Greenhouse-Geisser correction). (B) Averaged emotional intensity scores from 59 participants in Experiment 1. No significant difference is found among emotional conditions. (C) Averaged valence scores from 31 participants in Experiment 2. A significant main effect of emotional conditions is evident ($F_{1.5, 45.9} = 60.10$, $p < 0.001$, $\eta 2$ p = 0.667, Greenhouse-Geisser correction). (D) Averaged arousal scores from 31 participants in Experiment 2. A significant main effect of emotional conditions is detected ($F_{2, 29} = 9.92$, $p = 0.001$, $\eta 2$ p = 0.005). (*$p < 0.05$, **$p < 0.01$, ***$p < 0.001$).

Furthermore, an analysis of emotional arousal revealed that fearful scenes evoked the highest arousal levels (5.27 ± 0.24), followed by happy scenes (4.94 ± 0.25). In contrast, neutral scenes registered the lowest arousal levels (4.20 ± 0.25), as depicted in Fig 3D. A repeated-measures ANOVA was conducted, and Mauchly's Test of Sphericity was not significant ($p = 0.052$), indicating that the assumption of sphericity was met. The results demonstrated a significant main effect of emotional conditions on arousal ($F_{2, 29} = 9.92$, $p = 0.001$, $\eta 2$ p = 0.406). *Post hoc* tests using Bonferroni correction indicated significant arousal differences between fearful and neutral conditions ($p < 0.001$) and between happy and neutral conditions ($p = 0.001$). These findings suggest that emotional scenes triggered higher emotional responses than neutral scenes. Importantly, none of the participants reported prior knowledge of the Kuleshov effect.

### 3.2.2. fMRI data.    *Neutral face processing in each emotional condition*

In the Kuleshov sequence, viewers first see a neutral face, followed by an emotional scene, and then a similar neutral face. Following exposure to the fearful scene, viewers attribute emotion to the neutral face, as indicated by the bar representing the fearful condition in Fig 3C. This suggests a perceived new meaning for the second neutral face. Detecting neural correlates of this phenomenon is crucial for affirming the existence of the Kuleshov effect.

To uncover the neural correlates associated with the new meaning attributed to the second face, our fMRI analysis compared brain activity between Face_2 and Face_1. We hypothesized that Face_2, when contrasted with Face_1, would display heightened activation across the entire brain, with distinct regions being activated under different emotional conditions.

In summary, we observed widespread activation across the entire brain in each emotional condition, and the specific activated regions varied (Fig 4). In the fearful condition, subtracting Face_1 from Face_2 revealed 46 clusters, including bilateral activation in the hippocampus, insula, precentral gyrus, postcentral gyrus, ACC, posterior cingulate cortex (PCC), orbitofrontal cortex (OFC), inferior parietal lobe (IPL), and cuneus (for additional AAL atlas labels, refer to S1 Table and Fig 4B). These regions are implicated in emotional memory and attentional

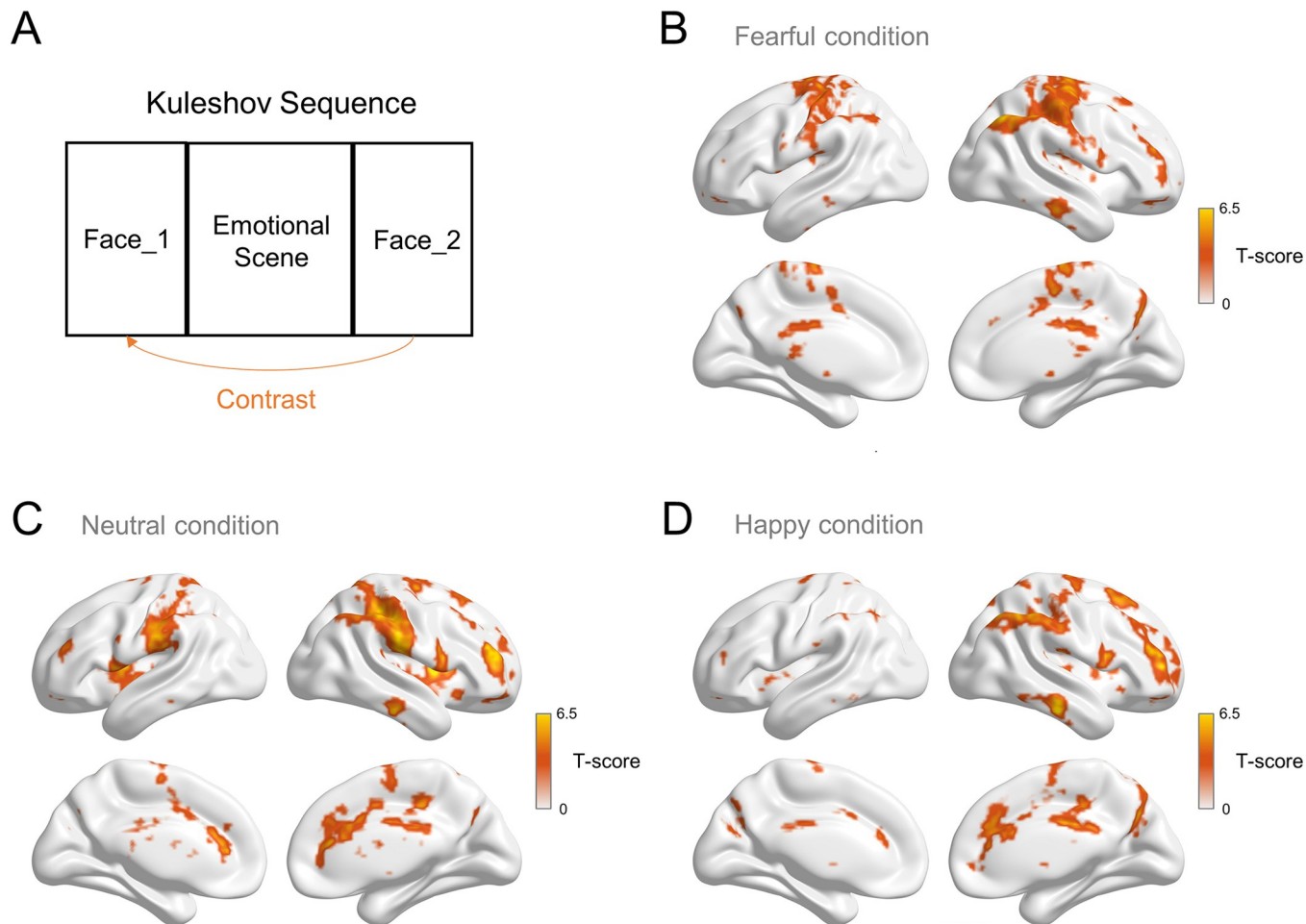

**Fig 4. Comparison of brain activity between Face_2 and Face_1 in each emotional condition.** Brain activity was obtained by subtracting Face_1 from Face_2 (A), and the contrasts were conducted for the fearful condition (B), neutral condition (C), and happy condition (D). ($p < 0.05$, FDR-corrected, cluster size > 5 voxels).

control, crucial for the Kuleshov effect in fearful conditions. Similarly, in the happy condition, subtracting Face_1 from Face_2 identified 51 clusters with bilateral activation in the insula, middle frontal gyrus (MFG), ACC, midcingulate cortex (MCC), PCC, IPL, and precuneus (for additional AAL atlas labels, refer to S2 Table and Fig 4D). These activated regions underscore the role of emotional memory and attentional control in the Kuleshov effect during happy conditions. In the neutral condition, subtracting Face_1 from Face_2 revealed 55 clusters with bilateral activation in the supplementary motor area (SMA), postcentral gyrus, precentral gyrus, IPL, MFG, ACC, and PCC (for additional AAL atlas labels, refer to S3 Table and Fig 4C). These activated regions suggest the involvement of motor control and sensory information processing in the Kuleshov effect during neutral conditions.

### Interaction between neutral face processing and emotional conditions

After viewing the Kuleshov sequence, participants attribute negative emotions to neutral faces in fearful conditions and positive emotions in happy conditions. This suggests an interaction between neutral facial processing and emotional conditions. This dynamic interaction adds a crucial layer to our comprehension of the Kuleshov effect and its intricate relationship with context-dependent biases.

Examining the (Face_2 −Face_1) contrast in the fearful condition against the neutral condition revealed activation in 10 clusters, including the right insula, bilateral precentral gyrus, bilateral postcentral gyrus, right SMA, and bilateral paracentral lobule (Table 1 and Fig 5A). These specific brain regions provide insights into the neural substrates of the Kuleshov effect in fearful scenes.

Similarly, analyzing the (Face_2 −Face_1) contrast in the happy condition against the neutral condition uncovered 10 clusters associated with happy scenes (Table 1 and Fig 5B). These regions, including the right hippocampus, bilateral PHC, bilateral AG, right MFG, right superior frontal gyrus (SFG), left IPL, right superior parietal gyrus (SPG), bilateral FG, bilateral retrosplenial cortex (RSC), left inferior temporal gyrus (ITG), left IPL, and right middle temporal gyrus (MTG), strongly indicate the presence of the Kuleshov effect in happy scenes. The activation patterns suggest heightened emotional memory engagement and increased attention to scene content in response to happiness, unveiling distinct brain regions that generate the Kuleshov effect within happy scenes.

## 4. Discussion

Lev Kuleshov conducted the seminal Kuleshov experiment nearly a century ago, which featured a face-scene-face sequence [12]. The experiment provided foundational support for the original montage theory and was a pioneering effort to explore the impact of film elements on viewers' perception within the realm of film cognition. The Kuleshov structure also utilized contemporary POV editing techniques. From both classical theoretical evidence and contemporary film cognition perspectives, revisiting the Kuleshov effect not only reexamines its existence but also highlights the efficacy of contemporary film editing practices. This study aims to confirm the existence of the Kuleshov effect through behavioral and fMRI experiments.

### 4.1 Enhancing ecological validity of the Kuleshov effect with authentic films

As the Kuleshov effect is a film editing phenomenon, validating this effect should ideally involve continuous films, rather than isolated images or clips. Earlier investigations [13,19,20] relied on facial expressions from the KDEF picture set and utilized zoom-in techniques to simulate dynamic faces from static images. These efforts faced challenges in achieving realism, including subtle facial tremors induced by individuals remaining still, the natural occurrence

**Table 1. fMRI Results: Interaction between neutral face processing and emotion conditions.**

| Brain Region | AAL Atlas Labels | Peak Voxel Coordinate (MNI) | Cluster Size (KE) | T-score |
|---|---|---|---|---|
| *Fearful [Face_2 –Face_1] > Neutral [Face_2 –Face_1] (FDR-corrected cluster threshold, p < 0.05)* | | | | |
| Right Insula/Rolandic Operculum | Insula_R<br>Rolandic_Oper_R | 36, -18, 18 | 6 | 4.813 |
| Left Precentral Gyrus | Precentral_L | -24, -24, 54 | 44 | 6.032 |
| Right Precentral Gyrus | Precentral_R<br>Postcentral_R | 28, -24, 54 | 34 | 6.200 |
| Left Postcentral Gyrus/Precentral Gyrus | Postcentral_L<br>Precentral_L | -38, -26, 56 | 5 | 5.242 |
| Left Postcentral Gyrus | Postcentral_L | -48, -14, 58 | 5 | 5.323 |
| Left Paracentral Lobule | Paracentral_Lobule_L | -6, -30, 70 | 20 | 5.355 |
| Right Paracentral Lobule | Paracentral_Lobule_R | 10, -30, 68 | 16 | 5.611 |
| Right SMA | Supp_Motor_Area_R | 6, -20, 68 | 8 | 5.276 |
| Right Paracentral Lobule/Postcentral Gyrus | Paracentral_Lobule_R<br>Postcentral_R | 12, -34, 80 | 14 | 5.550 |
| Left Paracentral Lobule | Paracentral_Lobule_L<br>Precentral_L | -8, -20, 80 | 52 | 6.999 |
| *Happy [Face_2 –Face_1] > Neutral [Face_2 –Face_1] (FDR-corrected cluster threshold, p < 0.05)* | | | | |
| Right Limbic Lobe | Fusiform_R<br>ParaHippocampal_R<br>Lingual_R<br>Cerebellum_4_5_R<br>Hippocampus_R | 32, -36, -12 | 244 | 8.713 |
| Left Limbic Lobe | Fusiform_L<br>ParaHippocampal_L<br>Lingual_L<br>Cerebellum_4_5_L<br>Temporal_Inf_L | -30, -40, -10 | 197 | 7.027 |
| Left Retrosplenial Cortex | Calcarine_L<br>Cuneus_L<br>Precuneus_L<br>Lingual_L<br>Occipital_Sup_L<br>Vermis_4_5 | -18, -58, 14 | 150 | 6.375 |
| Right Retrosplenial Cortex | Precuneus_R<br>Calcarine_R<br>Lingual_R<br>Cuneus_R | 18, -54, 20 | 247 | 7.490 |
| Left Occipital-Parietal Cortex | Occipital_Mid_L<br>Angular_L<br>Parietal_Inf_L | -32, -84, 30 | 338 | 6.712 |
| Right Occipital-Temporal Cortex | Occipital_Mid_R<br>Angular_R<br>Occipital_Sup_R<br>Temporal_Mid_R | 36, -78, 38 | 273 | 6.092 |
| Right Precuneus | Precuneus_R<br>Parietal_Sup_R | 12, -70, 52 | 61 | 4.625 |
| Right Angular Gyrus | Angular_R | 42, -60, 50 | 9 | 4.350 |
| Right Superior Frontal Sulcus | Frontal_Sup_2_R<br>Frontal_Mid_2_R | 28, 12, 54 | 9 | 4.495 |
| Right Superior Frontal Sulcus | Frontal_Sup_2_R<br>Frontal_Mid_2_R | 32, 14, 62 | 19 | 4.424 |

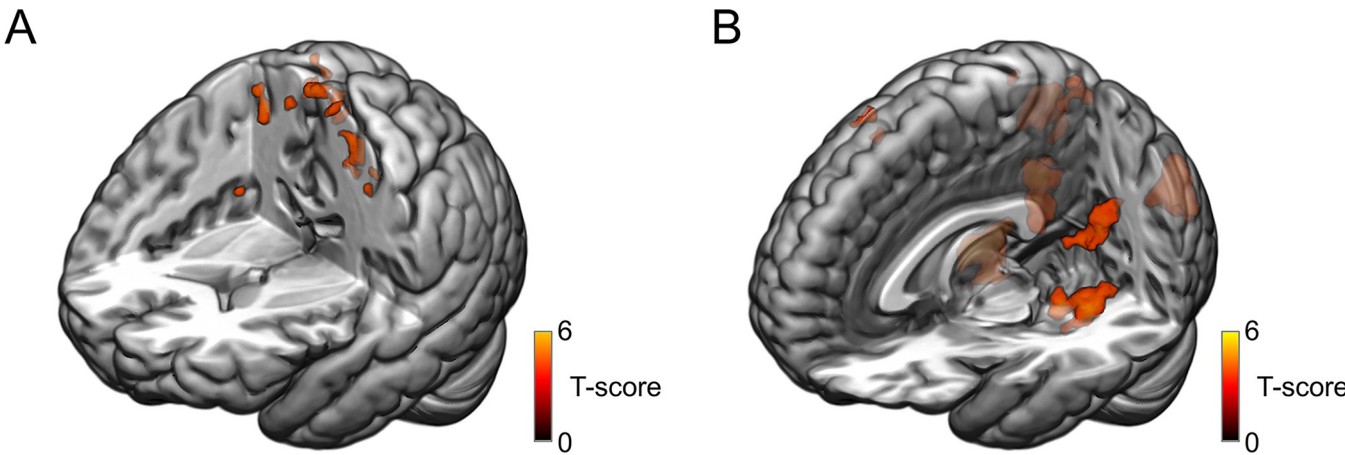

**Fig 5. Interaction between neutral face processing and emotional conditions.** (A) Brain regions showing the (Face_2 –Face_1) contrast in fearful condition minus the (Face_2 –Face_1) contrast in neutral condition. (B) Brain regions showing the (Face_2 –Face_1) contrast in happy condition minus the (Face_2 – Face_1) contrast in neutral condition ($p < 0.05$, FDR-corrected, cluster size > 5 voxels).

of eye blinking during observations, and the static background [22,23,35], making the simulation of static images as real-life films difficult. Moreover, earlier investigations selected face and emotional scenes separately rather than matching them as an integrated film [13,19]. Since the films closely mirror reality [36,37], adherence to continuity-editing rules aids viewers in perceiving the film as an emulation of the real world [7,38,39]. Consequently, using isolated neutral faces and emotional scenes separately, without considering the coherence of the backgrounds, led to spatial discontinuities.

To address these challenges, we employed a meticulous approach to capture neutral facial expressions against a blue background and filmed emotional scenes, along with their surrounding environment, in high resolution. Subsequently, we replaced the background with surrounding images or videos (S1 Fig and S1–S9 Videos), aiming for seamless integration of neutral faces into the film's context [39]. These rigorous methodologies guarantee a thorough reexamination of the Kuleshov effect in an authentic film scenario, eliminating potential confounding factors arising from neutral faces in the observation of the Kuleshov effect bias.

## 4.2 Confirming the existence of the Kuleshov effect through behavioral experiments

While the Kuleshov effect stands as a cornerstone in film studies, the early evidence from the original Kuleshov experiment is often considered ambiguous due to its limited experimental controls and the subjective assessments of the viewers' responses. It lacks clarity [12], leaving the existence of the Kuleshov effect an open question. Over the past three decades, despite replicated experiments suggesting the detection of the Kuleshov effect [13,17,19,20], many relied on inauthentic film materials, such as static images, which may not accurately capture the dynamic nature of actual film viewing. Consequently, their findings may not generalize to the existence of the Kuleshov effect in an authentic film scenario.

In the current study, we utilized authentic film materials (S1 Fig) and conducted two experiments using the face-scene-face Kuleshov sequence (Figs 1B and 2). Although a rating experiment established a baseline for neutral faces, participants were asked to rate the valence and the emotional intensity/arousal of the neutral face after watching the face-scene-face sequence, revealing that the emotional scene indeed affects the perception of emotion from a neutral face, whether viewed on a laptop (Fig 3A) or in an MRI scanner (Fig 3C). The ANOVA results

showed a significant difference in valence between emotional and neutral conditions, suggesting that a viewer's interpretation of an actor's expression is significantly influenced by the emotional scene, leading to a context-dependent bias in emotional perception. According to contextual framing [10,40], this context-dependent bias indicates that when viewers resolve ambiguous facial emotions, they rely on external cues, such as the emotional context—in this case, the emotional scenes in the current study. The emotional context influences the judgment of neutral faces in our study, supporting the Kuleshov effect.

Notably, the Kuleshov sequence triggered higher arousal in the fearful condition compared with the neutral condition, consistent with prior research [19]. Additionally, arousal levels in the happy condition were also higher than in the neutral condition. These observations suggest that the emotional conditions initiate a context-dependent bias and intensify viewers' emotional experiences. Effective control of the influence of prior knowledge about the Kuleshov effect ensures the reliability of our experimental results, as depicted in (S5 Fig). In line with previous studies [19,20], the behavioral results from Experiments 1 and 2 collectively support the existence of the Kuleshov effect in an authentic film scenario.

## 4.3 Dissociation between emotional intensity rating and physiological experience of arousal rating

In our study, while emotional scenes influenced arousal ratings, they did not similarly affect ratings of emotional intensity. This distinction underscores the Cognitive Appraisal Theory [31], which posits that individual appraisals of emotional stimuli significantly influence emotional experiences but do not always correspond with physiological reactions. Emotional intensity refers to the perceived emotional degree in the actor's performance, whereas arousal measures the viewer's physiological response to the neutral face. In Experiment 1, the influence of emotional scenes on emotional intensity was not significant ($F_{2, 57} = 0.13$, $p = 0.879$, as shown in Fig 3B), suggesting that participants utilized external appraisal methods and perceived similar levels of emotional intensity in the actors' performances across different emotional scenes. However, an analysis of emotional arousal in Experiment 2 revealed distinct variations across different types of emotional scenes ($F_{2, 29} = 9.92$, $p = 0.001$, as shown in Fig 3D). Fearful scenes evoked the highest levels of arousal (5.27 ± 0.24), followed by happy scenes (4.94 ± 0.25), while neutral scenes registered the lowest (4.20 ± 0.25). These findings illustrate that emotional scenes (fearful and happy) can trigger heightened physiological responses compared to neutral scenes, thereby highlighting the role of cognitive appraisal in the differential emotional processing of stimuli. For example, in cognitive appraisal, fearful scenes lead to more threat perception than neutral scenes, resulting in higher arousal ratings. The divergence in findings between the two experiments reflects the distinct tasks employed: Experiment 1's focus on perceived emotional intensity and Experiment 2's measurement of physiological arousal. This demonstrates the nuanced ways viewers respond to cinematic stimuli, as cognitive appraisal may influence emotional intensity ratings, while physiological arousal may be more directly responsive to the emotional content of scenes.

## 4.4 Inferring actors' emotions under POV structure

Through the POV sequence in the current experiment, we discovered that the POV structure significantly influences how viewers perceive emotions from actors (Fig 3A and 3C). Beyond the contextual framing—where the emotional scene serves as a backdrop that shapes emotional perception [41]—another pivotal factor is the POV structure's ability to prompt viewers to infer actors' emotions directly. This sequence in our study includes a glance shot, an object shot, and a subsequent return to the glance shot. The glance shot, a close-up, focuses the viewer

on the actor's facial expression, enhancing engagement and emotional attribution. These close-up shots reveal the actor's deictic gaze, fostering social cognition and theory-of-mind in viewers [42–45], thus enabling them to attribute nuanced mental and emotional states to the actors. As viewers hypothesize about the actors' thoughts, the ensuing object shot—aligned near the 180-degree axis—visually conveys what the actors are presumably observing from their egocentric viewpoint [30], bridging the viewer's understanding of the scene's context. This method of editing not only deepens the emotional connection but also transitions the viewer from an external observer to a more immersed perspective, aligning with the actor's viewpoint.

Moreover, research confirms that close-up shots are pivotal in aiding viewers to gauge an actor's emotional state effectively. For example, Balint [46] demonstrated that increasing the frequency of close-ups significantly boosts mental state attribution and viewer engagement, a finding that supports the argument for the power of editing in enhancing emotional perception. In our research, the Kuleshov sequence incorporated two such close-up shots, which proved instrumental in helping viewers infer the actors' emotional states more accurately. Although we consistently utilized a fixed camera angle to capture the actors' neutral expressions, the influence of camera positioning in the film remains a critical factor. Studies, such as those by Clinton(2017) [3], illustrate that variations in camera angles, such as low-angle and high-angle shots, distinctly influence viewers' emotional interpretations. The exploration of how these angles affect emotional perception will be a primary focus of our future research, aiming to further delineate the nuanced interplay between film technique and viewer emotional response.

## 4.5 Confirming the existence of the Kuleshov effect through fMRI

The Kuleshov effect encompasses two primary interpretations. The first is that viewers perceive a new meaning, such as emotion from a neutral face, as illustrated by the first bar in Fig 3A. The second interpretation is a context-dependent bias where, compared to a neutral condition, viewers perceive significant negative or positive emotions in emotional conditions, as demonstrated by the ANOVA results of Fig 3A. To confirm the existence of the Kuleshov effect through fMRI, these interpretations were examined through contrasting analyses.

In exploring the perceived new meaning of the second neutral face (Face_2), we investigated the neural correlates associated with this interpretation. The processing of the neutral face involved contrasting the activation in Face_2 with Face_1, revealing multiple regions of activation (Fig 4 and S1–S3 Tables). Specifically, as a data exploratory study, our examination of the fearful condition revealed activations in various brain regions, including the insula, cuneus, ACC, PCC, IPL, hippocampus, and OFC (Fig 4B). In contrast, the happy condition (Fig 4D) displayed distinct activation patterns compared to the fearful condition, indicating the diversity in the creation of new meaning under different emotional contexts.

To understand why these regions showed activation, we rely on the idea that engaging in movie viewing captures the complexities of real-world interactions, allowing viewers to immerse themselves in the film [36]. Thus, the Kuleshov sequence leads viewers to perceive 'more emotions' in Face_2 than in Face_1, indicating that viewers engage in the theory of mind and emotion generation. Viewers generate emotions and use the theory of mind to assume the actor experiences the same emotions. Regarding emotion, the activation of the insula—a region associated with emotion processing—is noteworthy [47]. As the activation of the ACC, PCC, and IPL was also observed, it suggests that viewers employ the theory of mind to interpret the actor's feelings [48,49]. The rationale behind emotion perception from a neutral face could be attributed to contextual processing [41]. Zheng et al. (2022) [41], using

intracranial electroencephalogram recordings, investigated context-specific modulation among the amygdala, hippocampus, and OFC. They found that the OFC modulates the hippocampus and amygdala when perceiving emotions on a neutral face. The activation of the hippocampus and OFC in this study aligns with Zheng's findings, supporting the notion that contextual processing is a likely mechanism underlying the Kuleshov effect.

For the context-dependent bias, we initially examined activations of Face_2 in fearful and happy conditions compared to neutral conditions (S2 Text and S7 Fig), revealing distinct brain activation patterns in neutral faces when they are preceded by fearful or happy scenes. Subsequently, we conducted a comparative analysis by contrasting the activation differences between Face_2 and Face_1 in the emotional conditions with a similar contrast in the neutral condition, thereby revealing the interaction between neutral face processing and emotional conditions (Fig 5).

Several brain regions were identified as active in the contrast of Face_2 among emotional conditions (S7 Fig and S4 Table). Contrasting Face_2 between fearful and neutral conditions revealed activations in PCC, AG, and FG. The involvement of the PCC and AG, vital components of the default mode network (DMN), reinforces their role in contextual framing, particularly during the interpretation of the Kuleshov effect [50,51]. Additionally, the activation of the PCC, as noted in studies on emotional word processing [52], further supports its role in evaluating emotional meanings. The activations observed in the AG and FG during fearful conditions align with findings by Quinones Sanchez et al. [53], emphasizing heightened involvement in facial recognition and memory retrieval in fear-inducing circumstances. Contrasting Face_2 between happy and neutral conditions revealed activations in the right cuneus, precuneus, and CAL. The correlation between gray matter volume in the right precuneus and subjective happiness, as demonstrated by Sato [54], underscores the role of the precuneus in our activation results for perceiving happiness on neutral faces.

Significant brain areas were observed in the interaction between neutral face processing and emotional conditions. For the fearful contrast, the activation in sensory and motion-related areas such as the Insula, SMA, and premotor cortex may indicate a fight-or-flight response [55], as these regions are typically involved in processing environmental threats and preparing the body for rapid action, such as reacting to a fearful film sequence. For the happy contrast, activations in the limbic system areas indicate that the brain's reward pathways are engaged [56], reflecting the reward system in response to a happy film sequence that elicits pleasure.

In summary, the first interpretation of the Kuleshov effect is supported by neural results that confirm the Kuleshov sequence indeed creates new meaning for the second neutral face, supporting the montage theory that '1 + 1 > 2'. For the second interpretation, network-level brain systems were involved, generating distinct emotions after viewers watched varied film sequences. Collectively, these findings confirm the existence of the Kuleshov effect at the neural level.

## 4.6 Comparative neurocinematic methodologies in current research

Film is a complex art form, and adopting neurocinematic approaches to explore its impact on viewer psychology is essential. Currently, neurocinematics predominantly utilizes two main research methodologies: The experimental Approach and the Naturalistic Approach.

In the Experimental Approach, researchers design experiments to test specific hypotheses, such as how simple scene setups affect viewers' emotional responses [10,57]. Using the GLM method, this approach effectively investigates the mechanisms through which specific film elements impact audiences. However, the ecological validity of the materials used often does not encompass all types of films, limiting the generalizability of the findings.

In the Naturalistic Approach, researchers analyze large datasets to identify patterns and correlations under less controlled, more natural conditions [27]. Methods like ICA and ISC are used to manage complex scene data from films. For instance, Uri Hasson demonstrated the consistency of brain activation across viewers in response to emotional films [27]. Janne Kauttonen explored correlations between camera shots, actor movements, and brain responses in non-narrative films [58]. Meanwhile, Gal Raz examined changes in brain networks and dynamic brain responses to complex film scenes [59,60]. Although the complex mise-en-scène in naturalistic films limits the precise neural mapping of film elements, their research contributes to understanding the relationship between film elements and neural responses, further grounding the field of neurocinematics.

These Naturalistic Approach studies have identified some common brain areas, such as the Insula-based network [60], which align with the activation results in the current study using the Experimental Approach. The complexity of the scenes often hampers the precise isolation of the independent effects of various film elements on viewer impact. Although this study employs the GLM method to explore these effects, integrating the advantages of the GLM method with the ISC method represents a promising direction for future research. This integrated approach could offer a more comprehensive method to analyze and understand how films influence viewers' emotions and cognition through diverse film elements and brain network activities.

## 4.7 Limitations and future directions

The current study, while confirming the existence of the Kuleshov effect with authentic films, encounters several limitations due to the multifaceted nature of psychological and cinematic responses and the varied socio-cultural contexts of the participants.

Firstly, it is crucial to acknowledge the potential impact of various factors on observing the Kuleshov effect, such as viewer gender, film-watching experience [18], and cultural background. While eliminating these influences remains challenging, we carefully balanced participants' gender in our experiments and conducted a comprehensive questionnaire analysis to assess participant's film-watching frequency statistically. The results, depicted in (S6 Fig), indicate that watching frequency did not significantly affect the observation of the Kuleshov effect. Additionally, considering the global nature of film watching [61], there could be a cultural background and cross-race effects in film perception [62]. Thus, exploring the Kuleshov effect with similar film sequences across diverse cultures remains an important question for future investigation. In light of these considerations, our study aims to reexamine the existence of the Kuleshov effect within a Chinese population. Drawing on Anna Kolesnikov's analysis of the effect's geocultural origins [63], which validated the effect in Europe, the US, and the USSR, we explore how different cultural interpretations influence the perception and implementation of this film theory in China. This research contributes to the broader discourse by empirically demonstrating the applicability of the Kuleshov effect in Chinese culture. Our study also enriches our understanding of cross-cultural cinematic psychology.

Secondly, the results from the current study may not fully reflect commercial fiction films. In such films, actors typically express a range of emotions within the POV structure [29], and sound plays a critical role in creating an authentic film experience, potentially contributing to what might be termed the auditory Kuleshov effect [64]. Despite our intentional removal of all sound from the film sequences, in line with a previous study [19], the potential interaction between visual and auditory Kuleshov effects and the manipulation of different facial expressions warrant further exploration. This exploration could help ascertain the extent to which our findings can be generalized to typical commercial viewing experiences.

Thirdly, we must acknowledge the limitations of our tools for subjective emotion measurement. Emotions are complex and diverse, yet we have measured emotional perception from only three aspects: valence, emotional intensity, and arousal. Other dimensions such as categorization and dominance also exist, where overlaps between these emotional measurement tools occur [19]. Regarding the form of the tools, this study employed VAS to facilitate participants' understanding of emotions. Although VAS has been validated [34], whether participants truly grasped the emotions through VAS remains to be explored. Concerning the range of the scales, this study used a smaller range in Experiment 1 and a larger range in Experiment 2. Although both experiments found similar statistical differences between emotional conditions with valence scales, as evidenced by the p-values and effect sizes, this indicated a robust observation of the Kuleshov effect across different ranges of scales. However, a smaller range can force choices for participants but may also overlook individual differences. In contrast, a larger range can capture individual responses and enhance the sensitivity of measurements by reducing the potential for ceiling or floor effects. Regarding participant characteristics, it should be noted that cultural backgrounds influence individual responses [65,66], which might further affect the emotional scoring tendencies in the current study. For example, Western individuals might give overly positive responses, reducing discrimination in the lower range, while Eastern individuals may underutilize the upper end of the scale. Additionally, Western participants might approach multidimensional scales analytically, whereas Eastern participants might approach them holistically. Thus, considering that our results stem from only three dimensions of emotion using VAS scales, future research should explore additional emotional aspects, employ a broader range of scales, and consider the impact of cultural backgrounds on participants' responses to these scales. This approach will enable a more effective reexamination of the Kuleshov effect across different emotional dimensions.

Fourthly, while the amygdala's involvement in fMRI studies of the Kuleshov effect has been documented [10], we did not detect its activation with FDR correction in our study. This disparity may arise from paradigm differences, as Mobbs's study employed a scene-face sequence with static images, while ours incorporated a face-scene-face sequence with authentic film clips. Given the practical challenges in creating authentic films, we were constrained in capturing a limited number of scenes and conducting a restricted number of trials in the fMRI experiment. Future research should consider expanding the number of trials to further observe the activation patterns associated with the Kuleshov effect.

Lastly, in the timeline of a single trial in Experiment 2, the activation of Face_2 occurs earlier than the valence rating, indicating a disconnect between unconscious physiological activation and conscious emotional perception. Therefore, physiological responses represent immediate activation, while subjective reports reflect the participants' systematic assessment of emotional perception. This timing difference is crucial for interpreting the Kuleshov effect, as it suggests that physiological and subjective responses are part of a sequential process, rather than simultaneous phenomena. Despite these differences in the timing of measurements, both the localized brain activation patterns and the subsequent behavioral responses reveal the presence of the Kuleshov effect. Although understanding how the Kuleshov effect is generated—specifically, how brain responses transition to behavioral responses and result in complex emotional perception—is critical, a detailed exploration of this transition can further elucidate the underlying mechanisms of emotional processing in film viewing. It is beyond the scope of this study to fully explore this transition. Given the complexity of the brain-behavior relationship [67], future research addressing this question may involve finite impulse response (FIR) time-course analysis within regions of interest (ROIs) or correlation analyses between ROIs and behavioral responses to help uncover the mechanisms behind the Kuleshov effect [68].

This will significantly enhance our understanding of how different types of responses collectively contribute to the interpretation of cinematic effects.

## 5. Conclusion

In this study, our main objective was to investigate the existence of the Kuleshov effect using authentic films within the realm of film cognition. We produced these film sequences and conducted two experiments. The results from the behavioral experiments indicate that viewers perceive emotions on neutral faces after watching the Kuleshov sequence, demonstrating that film editing can alter viewer perception by introducing new meanings and thus confirming the existence of the Kuleshov effect through subjective measurement. Through a comparative analysis of the valence of neutral faces across different emotional conditions, we identified a context-dependent bias in the Kuleshov effect. Specifically, viewers tend to perceive negative emotions on neutral faces after watching fearful film sequences and positive emotions after watching happy film sequences. Furthermore, our exploration extended to the neurobiological basis of the observed phenomena through fMRI experiments. The activation patterns in the insula, cuneus, precuneus, hippocampus, PCC, PHC, FG, and OFC not only support the existence of the Kuleshov effect from an objective measurement standpoint but also suggest that contextual framing is the mechanism underlying the Kuleshov effect. These findings provide robust evidence supporting the existence of the Kuleshov effect at both behavioral and neural levels. Our study aligns with research utilizing the Kuleshov paradigm to investigate contextual framing [10,19,28,41], offering additional evidence that film editing can significantly influence viewer's perceptions, evoking diverse emotional responses. Furthermore, our study contributes valuable insights into the POV structure and neurocinematics, advancing the knowledge of film cognition.

## Supporting information

**S1 Text. Materials rating experiment.** The detailed methods of materials rating experiment.
(DOCX)

**S2 Text. fMRI contrasting results.** Comparison of brain activity for Face_2 between emotional and neutral conditions.
(DOCX)

**S1 Fig. Film clip shooting process.** This figure illustrates the process of shooting film clips. We employed a cinecamera, lighting equipment, and a blue screen for shooting the film clips (A). Filming encompassed the sequential capture of a neutral face and an emotional scene within a face-scene-face sequence employing the shot-reverse-shot structure. The shot of the neutral face was taken with the camera positioned close to the 180-degree axis, facing the face (B). The actor was directed to look at a cross marker before them (C). As for the reverse shot of the emotional scene, the camera was placed near the 180-degree axis, oriented toward the object (D) and (E).
(TIF)

**S2 Fig. Examples of emotional scenes.** These scenarios spanned various genres, including horror, documentary, and comedy. Fearful scenes depicted chilling scenarios such as peeping, murder, and a ghost. Neutral scenes featured commonplace objects like a printer, a bus stop, and a cup. Happy scenes showcased joyful images of an attractive girl, food, and humorous expressions.
(TIF)

**S3 Fig. Film material production steps.** The current figure outlines the steps involved in creating the film clips. The process consists of three main steps for generating the neutral face material. First, we recorded a video featuring a facial shot without expression. Second, the background was substituted with a surrounding video or an image, all of which was shot along with an emotional scene. And, adjustments were made to the color temperature, brightness, and contrast of the facial videos to harmonize them with the emotional scenes. Lastly, we converted the color facial video into black and white [19]. Regarding the reverse shot, namely the emotional scene, we simply applied a black-and-white filter to the footage.
(TIF)

**S4 Fig. Comparison of high-authentic film clips and low-authentic film clips.** This figure compares highly authentic film clips and those with low ecological validity. Previous studies utilized static images for neutral faces [19] or used unmatched backgrounds with reverse shots [18]. These approaches significantly differ from authentic films. In contrast to the lower authentic film clips displayed in (A), neutral faces with replaced backgrounds (B) provide a more authentic and immersive experience. Therefore, there is a compelling need to reevaluate the Kuleshov effect using highly authentic films.
(TIF)

**S5 Fig. Behavioral results with subgroups of knowing the Kuleshov effect in experiment 1.** To identify and exclude participants with prior knowledge of the Kuleshov effect, we administered a knowledge test. Out of the 59 participants, seven already knew about the Kuleshov effect, while 52 did not. An Independent Samples Mann-Whitney U Test conducted on valence ratings across these two groups (unknown, known) revealed no significant differences between the groups, either in the fearful condition, neutral condition, or happy condition (all $p > 0.1$). This suggests that prior knowledge of the Kuleshov effect does not affect the observation of the Kuleshov effect.
(TIF)

**S6 Fig. Behavioral results with subgroups based on film-watching frequency in Experiment 1.** To examine whether film-watching frequency influences the observation of the Kuleshov effect, we conducted a subgroup comparison based on watching frequency. Of the 59 participants, 42 watched 0–2 films per week, 16 watched 3–5 films per week, and only one participant watched 6–8 films per week. Although there is a tendency for the middle-watching group (3–5 films per week) to have lower valence than the less-watching group (0–2 films per week), an Independent Samples Mann-Whitney U Test conducted on valence ratings across the two groups (0–2 films per week, 3–5 films per week) revealed no significant differences between the groups, either in the fearful condition, neutral condition, or happy condition (all $p > 0.1$). This suggests that film-watching frequency does not significantly affect the perception of the Kuleshov effect.
(TIF)

**S7 Fig. Comparison of brain activity for Face_2 between emotional and neutral conditions.** (A) Brain activity was obtained by subtracting Face_2 in the neutral condition from Face_2 in the fearful condition. (B) Brain activity was obtained by subtracting Face_2 in the neutral condition from Face_2 in the happy condition. ($p < 0.05$, FDR-corrected, cluster size > 5 voxels).
(TIF)

**S1 Table. fMRI Results: Face_2 minus Face_1 in fearful condition.** To uncover the neural correlates associated with the new meaning attributed to the second face, our fMRI analysis compared brain activity between Face_2 and Face_1 in fearful condition. ($p < 0.05$, FDR-corrected, cluster size > 5 voxels).
(DOCX)

**S2 Table. fMRI Results: Face_2 minus Face_1 in happy condition.** To uncover the neural correlates associated with the new meaning attributed to the second face, our fMRI analysis compared brain activity between Face_2 and Face_1 in happy condition. ($p < 0.05$, FDR-corrected, cluster size $> 5$ voxels).
(DOCX)

**S3 Table. fMRI Results: Face_2 minus Face_1 in neutral condition.** To uncover the neural correlates associated with the new meaning attributed to the second face, our fMRI analysis compared brain activity between Face_2 and Face_1 in neutral condition. ($p < 0.05$, FDR-corrected, cluster size $> 5$ voxels).
(DOCX)

**S4 Table. fMRI Results: Face_2 in fearful or happy condition minus Face_2 in neutral condition.** To probe the neural correlates of this Kuleshov effect bias, focus on how neutral faces exhibit distinct brain activation patterns when preceded by fearful or happy scenes. ($p < 0.05$, FDR-corrected, cluster size $> 5$ voxels).
(DOCX)

**S1 Video. Fearful film sequence: Peeping.** The combination of a female neutral face and a peeping scene.
(MP4)

**S2 Video. Fearful film sequence: A ghost doll.** The combination of a male neutral face and a ghost doll scene.
(MP4)

**S3 Video. Fearful film sequence: Murder.** The combination of a female neutral face and a murder scene.
(MP4)

**S4 Video. Neutral film sequence: A printer.** The combination of a male neutral face and a printer scene.
(MP4)

**S5 Video. Neutral film sequence: Traffic light for pedestrian crossing.** The combination of a female neutral face and a traffic light scene.
(MP4)

**S6 Video. Neutral film sequence: Cleaning the blackboard.** The combination of a male neutral face and a cleaning of the blackboard scene.
(MP4)

**S7 Video. Happy film sequence: An attractive girl.** The combination of a male neutral face and an attractive girl scene.
(MP4)

**S8 Video. Happy film sequence: Humorous expressions.** The combination of a female neutral face and humorous expressions scene.
(MP4)

**S9 Video. Happy film sequence: Natural Flower.** The combination of a female neutral face and a natural flower scene.
(MP4)

## Acknowledgments

We express our gratitude to Professor Yong He from IDG/McGovern Institute at Beijing Normal University for the assistance in the data collection process. Special appreciation is extended to the Sichuan Film and Television University film crew for their essential support in shooting films. The authors would also like to thank Ran Li, Xiang Xiao, Yanlin Zhu, Dayi Zhou, and Xinya Huang for their support during the study. The manuscript had appeared online as a preprint [69].

## Author Contributions

**Conceptualization:** Zhengcao Cao, Yashu Wang.

**Data curation:** Zhengcao Cao, Zhichen Shi.

**Formal analysis:** Zhengcao Cao.

**Funding acquisition:** Yiwen Wang.

**Investigation:** Zhengcao Cao, Yashu Wang, Yapei Xie, Yiren Zhong.

**Methodology:** Zhengcao Cao, Yashu Wang, Yapei Xie.

**Resources:** Liangyu Wu.

**Software:** Zhengcao Cao, Yapei Xie.

**Supervision:** Yiwen Wang.

**Visualization:** Zhengcao Cao.

**Writing – original draft:** Zhengcao Cao.

**Writing – review & editing:** Yashu Wang, Yiwen Wang.

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
