## [Decision Letter · Decision Letter 0]

25 Apr 2024

PONE-D-23-41138Reexamining the Kuleshov effect: behavioral and neural evidence from authentic film experimentsPLOS ONE

Dear Dr. Wang,

Thank you for submitting your manuscript to PLOS ONE. After careful consideration, we feel that it has merit but does not fully meet PLOS ONE’s publication criteria as it currently stands. Therefore, we invite you to submit a revised version of the manuscript that addresses the points raised during the review process.

Editor comments. First, I have to apologise for the delay with your manuscript. Despite I asked numerous experts to review, I was able to secure only one referee. In general, it seems that although your topic, cinematic cognition, is of considerable general interest, expert researchers within this speciality field generally do not have expertise in brain imaging. As the journal guidelines mandate a minimum of two expert reviews to proceed further, and since we have received only one at present, I have two options: either to reject your manuscript or to offer you an opportunity to revise it based on the available review. I have opted for the latter. I would like to emphasise, at this point, that we cannot proceed further in the process without finding an additional reviewer with expertise in the neuroimaging field. I would therefore invite you to resubmit your work following a thorough revision and to include a response letter addressing each comment made by Reviewer 1. I would suggest that you provide me with six potential reviewers (including email correspondence) whom you know have the expertise to assess your work adequately (and are likely to review). I wish you good luck with your revisions.

We look forward to receiving your revised manuscript.

Kind regards,

Michael B. Steinborn, PhD

Section Editor

PLOS ONE

 [This study was financially sponsored through the Arts Project of 2019 National Social Science Fund of China (grant no.19BC041) and the Arts Project of 2023 National Social Science Fund of China (grant no.23ZD07). The authors thank them for financial support.].  

3. Please remove your figures from within your manuscript file, leaving only the individual TIFF/EPS image files, uploaded separately. These will be automatically included in the reviewers’ PDF.

Reviewers' comments:

Reviewer's Responses to Questions

**Comments to the Author**

1. Is the manuscript technically sound, and do the data support the conclusions?

Reviewer #1: Yes

2. Has the statistical analysis been performed appropriately and rigorously? 

Reviewer #1: Yes

3. Have the authors made all data underlying the findings in their manuscript fully available?

Reviewer #1: No

4. Is the manuscript presented in an intelligible fashion and written in standard English?

Reviewer #1: Yes

5. Review Comments to the Author

Reviewer #1: The presents study was conducted to test the Kuleshov effect. Kuleshov was a Russian filmmaker and theorist who famously argued that pairing shots of actors conveying a neutral expression with shots depicting different scene will induces inferences about the affective states of the person. There has been some debate on the power of the Kuleshov effect and some research suggesting his claims were overstated. The present study tested the Kuleshov effect in the context of behavioral and fMRI data. Participants viewed short film clips that depicted an actor expressing a neutral facial expression with shots intended to depict happy, fearful, or neutral context. In both experiments, emotional judgments were made regarding valence and intensity/arousal. There were clear differences in the valance judgments, but and arousal judgments (experiment 2). The fMRI data suggested that there were dissociable brain networks associated with the different conditions.

Evaluation

This study can contribute to the cognitive studies of fiction film. The experiments were well conducted. However, the literature review does not reflect contemporary thinking and research about editing practices and point of view shot sequencing (of which the Kuleshov effect is representative) and the comprehension of filmed events. Moreover, there is a burgeoning line this this research that focuses on grounding the cognitive experience of fiction film in the brain. This study will have a greater impact on the filed if the authors revise the manuscript to better reflect its relationship to the field. My review is directed at helping the authors better realize the relationship of this study to the field of the cognitive study of film and neurocinematics.

Comments

1. Characterizing modern theories of the editing as montage theory is not reflective of contemporary thinking. David Bordwell’s seminal book of the structure of fiction film and his well-used textbook (Bordwell, 1985; Bordwell, Thompson, & Smith, 2019) describe editing practices as “continuity editing”. While montage was originally used to convey editing practices, it is now used to refelct a specific practice of editing shots in manner that depicts passage of time (e.g., the training scene in the movie Rocky is an example of this). As such, those most likely to read this article will not see the review as reflecting contemporary perspectives of editing practices and their impact of the cognition of fiction film.

Bordwell, D. (1985). Narration in the Fiction Film. Madison, WI: University of Wisconsin Press.

Bordwell, D., Thompson, K., and Smith, J. (2019). Film Art: An Introduction. 12th ed. McGraw-Hill.

2. The Kuleshov experiment was seminal in informing editing practices that involving pairing close-up shots of characters with other scenes to help the viewer infer what characters are thinking and feeling. A contemporary way to characterize this editing practice is point-of-view shot sequencing. The controversy over Kuleshov’s claims had everything to do with the strong argument that actors didn’t need to act, and that the editing sequence alone could convey the emotion. However, it is almost always the case that actors are asked to convey emotional expressions in point of view shot sequencies. This doesn’t change the importance of this study, but more accurately conveying the Kuleshov effect and its relationship to the practices of creating fiction film will better position this study. The authors may want to discuss the relationship of this study to contemporary studies that have either explored the impact of point of view shot sequencing or the role of closeup shots in helping viewers infer the emotional states of characters.

Bálint, K, E., Blessing, J. N., and Rooney, B. (2020). “Shot Scale Matters: the Effect of Close-up Frequency on Mental State Attribution in Film Viewers.” Poetics 83: 101480. https://doi.org/10.1016/j.poetic.2020.101480.

Clinton, J. A., Briner, S. W., Sherrill, A. M., Ackerman, T., & Magliano, J. P. (2017). “The Role of Cinematic Techniques in Understanding Character Affect.” Scientific Study of Literature, 7 (2), 177-202. https://doi.org/10.1075/ssol.16019.cli

I do question whether the materials have ecological validity, given that actors rarely are directed to maintain neutral expressions in the context of point of view shots sequencing. I think the authors should temper their claims about this aspect of the materials given that they may not reflect the practices of commercial fiction films, despite the fact that a filmmaker was involved in their development.

4. Uri Hasson and Gal Raz have pioneered the field of neurocinematics. Additional Gal Raz as conducted several studies grounding the affective response to film in brain networks. I would also like to draw the authors attention to the Baltic Neurocine group created by Pia Tikka (here’s a link to the 2022 conference, https://balticneurocineconf2022.sched.com/). Discussing this study in the context of the field of neuroscinematics will help the authors connect to the right readership for the article, which will help improve impact.

5. The authors should consider discussing Bohn-Gettler’s Process Emotion Task (PET) Theory to help justify the study. The theory specifically discusses the relationship between affect and comprehension processes and emphasizes both valance and arousal as important dimensions. One caveat, however, is that the theory focuses on the affective experiences of the viewer/reader, not inferences about the emotions of characters.

Bohn-Gettler, Catherine M. 2019. “Getting a Grip: the PET Framework for Studying How Reader Emotions Influence Comprehension.” Discourse Processes 56 (5-6): 386–401. https://doi.org/10.1080/0163853X.2019.1611174.

6. I’m not sure that I understand the differences between the emotional intensity and arousal judgments for Experiments 1 and 2. Please explain and what was different in the arousal judgments that lead to them finding significant differences in Experiment 2.

7. I know that the authors make materials available, but do you also make the data available? This is an important open science practice. If the data are made available, this needs to be made clearer. I usually include a data and materials access section in the methods section that specifies how to access the data

Joe Magliano

6. PLOS authors have the option to publish the peer review history of their article (what does this mean?). If published, this will include your full peer review and any attached files.

Reviewer #1: **Yes: **Joseph P. Magliano

---

## [Author Response · Author response to Decision Letter 0]

13 May 2024

PONE-D-23-41138

Reexamining the Kuleshov effect: behavioral and neural evidence from authentic film experiments

PLOS ONE

Dear Dr. Wang,

Thank you for submitting your manuscript to PLOS ONE. After careful consideration, we feel that it has merit but does not fully meet PLOS ONE’s publication criteria as it currently stands. Therefore, we invite you to submit a revised version of the manuscript that addresses the points raised during the review process.

Editor comments. First, I have to apologise for the delay with your manuscript. Despite I asked numerous experts to review, I was able to secure only one referee. In general, it seems that although your topic, cinematic cognition, is of considerable general interest, expert researchers within this speciality field generally do not have expertise in brain imaging. As the journal guidelines mandate a minimum of two expert reviews to proceed further, and since we have received only one at present, I have two options: either to reject your manuscript or to offer you an opportunity to revise it based on the available review. I have opted for the latter. I would like to emphasise, at this point, that we cannot proceed further in the process without finding an additional reviewer with expertise in the neuroimaging field. I would therefore invite you to resubmit your work following a thorough revision and to include a response letter addressing each comment made by Reviewer 1. I would suggest that you provide me with six potential reviewers (including email correspondence) whom you know have the expertise to assess your work adequately (and are likely to review). I wish you good luck with your revisions.

We look forward to receiving your revised manuscript.

Kind regards,

Michael B. Steinborn, PhD

Section Editor

PLOS ONE

Response: We appreciate the positive comments from the editor. We have made revisions based on Reviewer #1's suggestions. The potential reviewers are listed in the Cover Letter, labeled as 'Suggested Reviewers'.

Response: We thank the journal staffs for the reminder about manuscript style. The format of the affiliation has been revised accordingly.

 [This study was financially sponsored through the Arts Project of 2019 National Social Science Fund of China (grant no.19BC041) and the Arts Project of 2023 National Social Science Fund of China (grant no.23ZD07). The authors thank them for financial support.]. 

Response: We thank the journal staffs for the reminder regarding financial disclosure. We have included the Role of Funder statement. Relevant edits are listed as follows:

On page 46:

“The Arts Project of 2019 National Social Science Fund of China (grant no.19BC041) had role in study design, data collection and analysis, decision to publish. The Arts Project of 2023 National Social Science Fund of China (grant no.23ZD07) had role in preparation of the manuscript. ”

3. Please remove your figures from within your manuscript file, leaving only the individual TIFF/EPS image files, uploaded separately. These will be automatically included in the reviewers’ PDF.

Response: We thank the journal staffs for the reminder about manuscript style. There are no inserted figures in the current manuscript file.

Reviewers' comments:

Reviewer's Responses to Questions

Comments to the Author

1. Is the manuscript technically sound, and do the data support the conclusions?

Reviewer #1: Yes

2. Has the statistical analysis been performed appropriately and rigorously?

Reviewer #1: Yes

3. Have the authors made all data underlying the findings in their manuscript fully available?

Reviewer #1: No

4. Is the manuscript presented in an intelligible fashion and written in standard English?

Reviewer #1: Yes

5. Review Comments to the Author

Reviewer #1: The presents study was conducted to test the Kuleshov effect. Kuleshov was a Russian filmmaker and theorist who famously argued that pairing shots of actors conveying a neutral expression with shots depicting different scene will induces inferences about the affective states of the person. There has been some debate on the power of the Kuleshov effect and some research suggesting his claims were overstated. The present study tested the Kuleshov effect in the context of behavioral and fMRI data. Participants viewed short film clips that depicted an actor expressing a neutral facial expression with shots intended to depict happy, fearful, or neutral context. In both experiments, emotional judgments were made regarding valence and intensity/arousal. There were clear differences in the valance judgments, but and arousal judgments (experiment 2). The fMRI data suggested that there were dissociable brain networks associated with the different conditions.

Evaluation

This study can contribute to the cognitive studies of fiction film. The experiments were well conducted. However, the literature review does not reflect contemporary thinking and research about editing practices and point of view shot sequencing (of which the Kuleshov effect is representative) and the comprehension of filmed events. Moreover, there is a burgeoning line this this research that focuses on grounding the cognitive experience of fiction film in the brain. This study will have a greater impact on the filed if the authors revise the manuscript to better reflect its relationship to the field. My review is directed at helping the authors better realize the relationship of this study to the field of the cognitive study of film and neurocinematics.

Response: We appreciate the positive comments from the reviewer. We have included contemporary thinking and research about editing practices and point-of-view shot sequencing in the Abstract, Introduction, Discussion, and Conclusion. Relevant edits are listed as follows:

Abstract, on page 2-3:

“Film cognition explores the influence of cinematic elements, such as editing and film color, on viewers' perception. The Kuleshov effect, a famous example of how editing influences viewers' emotional perception, was initially proposed to support montage theory through the Kuleshov experiment. This effect, which has since been recognized as a manifestation of point-of-view (POV) editing practices, posits that the emotional interpretation of neutral facial expressions is influenced by the accompanying emotional scene in a face-scene-face sequence. ”

“This research also contributes to a deeper understanding of the impact of film editing on viewers' emotional perception from the contemporary POV editing practices and neurocinematic perspective, advancing the knowledge of film cognition. ”

Conclusion, on page 43-44:

“In this study, our main objective was to investigate the existence of the Kuleshov effect using authentic films within the realm of film cognition. ”

“Furthermore, our study contributes valuable insights to the POV structure and neurocinematics, advancing the knowledge of film cognition. ”

Comments

1. Characterizing modern theories of the editing as montage theory is not reflective of contemporary thinking. David Bordwell’s seminal book of the structure of fiction film and his well-used textbook (Bordwell, 1985; Bordwell, Thompson, & Smith, 2019) describe editing practices as “continuity editing”. While montage was originally used to convey editing practices, it is now used to refelct a specific practice of editing shots in manner that depicts passage of time (e.g., the training scene in the movie Rocky is an example of this). As such, those most likely to read this article will not see the review as reflecting contemporary perspectives of editing practices and their impact of the cognition of fiction film.

Bordwell, D. (1985). Narration in the Fiction Film. Madison, WI: University of Wisconsin Press.

Bordwell, D., Thompson, K., and Smith, J. (2019). Film Art: An Introduction. 12th ed. McGraw-Hill.

Response: We appreciate the reviewer’s valuable suggestion. We have read Bordwell’s book and included a discussion on continuity editing in the Introduction section. The current study was conducted within the realm of film cognition, with the Kuleshov effect serving as a famous example of how film editing affects emotional perception.

The Kuleshov experiment utilized the Kuleshov sequence to support the initial montage theory, and Bordwell distinguishes between the montage sequence and the montage concept in his textbook (Bordwell & Thompson, 2008). As you suggested that the Kuleshov sequence is similar to the montage sequence (which depicts the passage of time), we have included a few sentences distinguishing these terms in the Introduction.

Relevant edits are as follows:

Introduction, on page 3:

“Film cognition explores the intersection of cinematic elements and viewer psychology, focusing on how features such as camera angles, editing techniques, and color influence audience perception (Bordwell, 1985; Clinton et al., 2017; Germeys & D’Ydewalle, 2007; İyilikci et al., 2023; Smith, 2012; Stadler, 2018; Yilmaz et al., 2023). Through behavioral and neuroimaging experiments (Mobbs et al., 2006; Sanz-Aznar et al., 2023; Shimamura et al., 2014), research in this field examines the impact of these filmic components on viewers' interpretation of film narratives, contributing to advancing film theory and practice (Smith, 2013).

Reflecting on film history, a notable example of how editing influences viewers' perception is the Kuleshov effect. Identified through an experiment conducted in the 1920s by Soviet filmmaker Lev Kuleshov (1899–1970), this study juxtaposed an image of Russian actor Ivan Mozhukin’s neutral face with various emotional contexts in a series of frames arranged in a face-scene-face sequence (Pudovkin, 1970). This sequence includes segments featuring an actor with a neutral expression, followed by an emotional scene, and concluding with a repetition of the actor's neutral expression. During this experiment, viewers were tasked with assessing the emotions portrayed by the actor's face. Despite the actor's facial expression remaining consistent, viewers frequently attributed different emotional states to the actor, deeply linked to the accompanying emotional scenes. For example, when the emotional scene depicts a dead girl lying in a cabin, viewers perceive fear in the neutral face; conversely, when the scene features an alluring woman reclining on a couch, viewers interpret happiness in the neutral face. ”

Introduction, on page 8:

“While the Kuleshov sequence is often interpreted as an example of classic montage editing, where the goal is to combine images to generate new meanings—for instance, combining a clip of Mozhukin’s neutral face with a clip of a bowl of soup may create the idea of hunger—the original Kuleshov experiment also illuminates another fundamental aspect of film editing: continuity editing. The Kuleshov sequences serve as early examples of continuity editing (Barratt et al., 2016; Bordwell & Thompson, 2008), which includes a shot-reverse-shot structure, aiming to maintain a coherent narrative space and time. Moreover, these sequences exemplify point-of-view (POV) editing practices. In the 'glance shot', a character is shown gazing toward an offscreen entity (in this instance, Mozhukin’s neutral face), whereas the 'object shot' captures the item being looked at (such as a dead girl in a cabin). The POV structure is easily comprehended by viewers because it mimics the natural human and primate tendency to follow the gaze of an intentional agent (Carroll & Russell, 1996). Thus, reexamining the Kuleshov effect, particularly how it employs the POV structure and generates new meanings from neutral faces, contributes not only to validating classic montage theory but also to revealing the efficacy of contemporary film editing practices.”

Distinguishing terms 

---

## [Decision Letter · Decision Letter 1]

7 Jun 2024

PONE-D-23-41138R1Reexamining the Kuleshov effect: behavioral and neural evidence from authentic film experimentsPLOS ONE

Dear Dr. Wang,

Thank you for submitting your manuscript to PLOS ONE. After careful consideration, we feel that it has merit but does not fully meet PLOS ONE’s publication criteria as it currently stands. Therefore, we invite you to submit a revised version of the manuscript that addresses the points raised during the review process. Editor comments: Fortunately, three reviewers agreed to comment on your manuscript. I ask you to revise the manuscript accordingly and to respond to each of the comments in a point-by-point manner.

We look forward to receiving your revised manuscript.

Kind regards,

Michael B. Steinborn, PhD

Section Editor

PLOS ONE

Reviewers' comments:

Reviewer's Responses to Questions

**Comments to the Author**

1. If the authors have adequately addressed your comments raised in a previous round of review and you feel that this manuscript is now acceptable for publication, you may indicate that here to bypass the “Comments to the Author” section, enter your conflict of interest statement in the “Confidential to Editor” section, and submit your "Accept" recommendation.

Reviewer #2: (No Response)

Reviewer #3: (No Response)

Reviewer #4: (No Response)

2. Is the manuscript technically sound, and do the data support the conclusions?

Reviewer #2: Yes

Reviewer #3: Yes

Reviewer #4: Yes

3. Has the statistical analysis been performed appropriately and rigorously? 

Reviewer #2: Yes

Reviewer #3: Yes

Reviewer #4: Yes

4. Have the authors made all data underlying the findings in their manuscript fully available?

Reviewer #2: Yes

Reviewer #3: Yes

Reviewer #4: Yes

5. Is the manuscript presented in an intelligible fashion and written in standard English?

Reviewer #2: Yes

Reviewer #3: Yes

Reviewer #4: Yes

6. Review Comments to the Author

Reviewer #2: Reexamining the Kuleshov Effect: Behavioral and Neural Evidence from Authentic Film Experiments

The current study addresses an exciting and significant topic, the Kuleshov effect, by integrating behavioral and neural evidence to explore the underlying neural mechanisms. The study's rigorous experimental design and logical framework are commendable. The results are noteworthy, potentially elucidating the behavioral and neural manifestations of the Kuleshov effect. However, several critical issues need to be addressed to enhance the study's impact.

Major Issues:

1. Statistical Issues:

a. For whole-brain activation, the authors used FDR correction, but the cluster size threshold was set at above 5, which is quite small. Please provide justification for the chosen cluster size criterion.

b. The authors employed ANOVA followed by Fisher's Least Significant Difference for multiple comparisons. This method necessitates correction for multiple tests.

2. Experimental Design: On page 17, the fMRI experiment was conducted with a different presentation sequence compared to the behavioral study, which used randomization for the three conditions (happy, neutral, fearful). Could this sequential difference lead to interactions between trials? Given that emotions can persist over time, how can we ensure that the emotions induced by one trial do not influence subsequent trials?

3. The introduction and page 23 mention that the fMRI study shows that including jitters did not diminish the Kuleshov effect. Please omit this sentence as it detracts from the core objective of the study, which is to investigate the neural mechanisms of the Kuleshov effect.

4. In the fMRI analysis, three types of contrasts were used to investigate the neural basis of the Kuleshov effect. The second contrast, comparing happy or fearful conditions to the neutral condition, is similar to the third contrast, which compares Face 2 to Face 1 with the neutral condition. It would be more logical to either remove the second contrast or place it in the supplementary materials. The second contrast does not provide significant support for the current hypothesis and complicates the interpretation.

5. Interpretation of Results: The author's interpretation of the whole-brain results seems somewhat far-fetched. My view is that the whole-brain results for the fearful condition, compared to the neutral condition, show activation in sensory and motion-related regions, possibly indicating the fight-or-flight response. In contrast, the happy condition, compared to the neutral condition, shows activation in the limbic system, likely related to the reward system.

Minor Issues:

6. In the abstract, when first mentioning POV editing, please use the full term.

7. In the whole-brain activation tables, report only the main brain region name for each cluster rather than all regions.

8. On page 24, in the fMRI data results section, use a results summary or conceptual results and title the subsection "Results" rather than describing the analysis process.

Reviewer #3: This is an intriguing study in which the author innovatively applies cutting-edge cognitive neuroscience technology to the field of film cognition. Utilizing rigorous experimental design based on authentic films, the study verifies the existence of the Kuleshov effect from both behavioral and neural levels. This research offers a fresh perspective for examination of the Kuleshov effect, and its findings significantly enhance our understanding of its cognitive neural mechanisms. Additionally, it holds considerable guiding significance for future integrative research. However, it might be improved notably if issues including but not limited to the following items are taken care of:

Q1. The English language quality and presentation of the manuscript fall within an average range. It is recommended that professional refinement be conducted to enhance readability.

For instance, the expression "Somebody et al. [*]" is rarely used at the beginning of sentences in the main text. Moreover, the authors should vary their expressions as needed.

Additionally, while the journal requires numerical citations within the main text, it is desirable to include the year in the main body when referencing same important studies, to allow readers to quickly grasp the timeline of the related research.

In fact, I am not quite sure what the purpose of the penultimate paragraph in the author's introduction is, and how it relates to the context. If possible, I would appreciate it if the author could express it more clearly.

The manuscript contains numerous verbose expressions. For example, the second paragraph in the results section of 3.1. Experiment 1 is overly detailed; the relevant information could be consolidated with post-hoc analyses. Furthermore, the caption of Figure 3 includes redundant statistical information that is already present in the main text.

The abbreviation "fMRI" first appears on line 133 (in R1_reviwer manuscript, ), yet its full name is not introduced until line 145.

Q2. The authors claim the existence of the Kuleshov effect is contentious and use it as a starting point for their research. However, in presenting the opposing views, they only reference a single, dated piece of literature (Ref 16, Prince S, 1992). Hence, it is unclear whether this controversy is confined to the period before the 21st century, or if it still persists. If the controversy mainly existed in the past and a consensus has largely been reached today, then the focus of this research might not be on addressing this dispute.

Q3. The authors state that the primary goals of Experiment 1 (Exp1) and Experiment 2 (Exp2) are to verify the existence of the effect from both behavioral and neural levels. However, it appears that Exp2 also includes behavioral tests that are essentially the same as those in Exp1. The logic behind this design arrangement needs clarification from the authors.

Additionally, regarding the behavioral tests, aside from the most crucial valence rating of neutral faces, the author measured Emotional Intensity in Exp1 and a different metric, Arousal, in Exp2. We hope the author can also explain the purpose behind this arrangement in the introduction.

Moreover, when presenting the results, although some of the results in Fig 3C&D are similar to the behavioral outcomes of Exp1, they seem misplaced in the "3.1. Experiment 1" section.

Q4. Despite the authors' claims that they utilized fMRI technology to test the existence of the Kuleshov effect (as mentioned in the last paragraph of the introduction), I believe its role is not limited to this aspect alone. Here, I would like to know whether, prior to conducting the experiments, the authors had any specific expectations for the neural results of the study based on related theories and empirical research from the perspective of the cognitive processing that associated brain regions or networks when the effect occurs, or if it was solely a data exploratory study.

Similarly, we believe that the discussion section of this study lacks depth, particularly in the discussion of the neural results. For example, on line 750, the cuneus is the result obtained from the condition subtraction. According to the principle of subtraction, the probability that this result is specific to the role of visual processing is relatively small. Conversely, it is also related to emotion, memory and verbal comprehension. The authors should interpret these brain regions or networks from the perspective of the possible cognitive processing mechanisms involved in the Kuleshov effect's production.

Q5. Regarding the method section for Exp1, the authors should directly state whether the experiment used a between-subjects design or a within-subjects design.

Additionally, I am unclear why the authors randomly divided the Neutral Face clips material into three groups beforehand. It seems that they only needed to make a random match when combining faces and emotional scenes. Of course, this is not a mistake.

Q6. In the MRI Data Acquisition section 2.2.3, there are some non-standard expressions: the matrix has no units, and the length and width of the voxel size do not correspond, with 200/64 ≈ 3mm. Another important information, the volume number, should be reported.

Regarding the preprocessing of fMRI, the authors mention that they performed Field Map correction. It should be clarified whether they collected field map data, and the relevant parameter information should be presented in section 2.2.3.

When creating the GLM model for the first-level analysis of task-based fMRI activation, were the individual head motion parameters considered?

Q7. When presenting the results of ANOVA on behavioral data, we hope that the authors can report the effect sizes of their analysis.

Additionally, for neuroimaging analysis results, we would like to confirm if there are acturally no any significant negative activations.

Reviewer #4: I read the present manuscript with much interest. Taking a cue from previous studies (e.g., Calbi et al., 2017; 2019) that investigated the Kuleshov effect, the authors took a further step to improve the ecological and cinematic validity of the paradigm by constructing dynamic film scenes thus making the spatial continuity between glance shots and object shots more linear and consistent. In a first experiment they basically validated stimuli and experimental paradigm, while in the second they introduced slight changes to render the paradigm suitable to an fMRI investigation. While I appreciate the methodological effort of the authors, there are several limitations and aspects to be clarified.

General comment: reading is tiring at times because there are many repetitions in the text, this lengthens an already very long text.

ABSTRACT

Authors should better describe participants’ task; writing that “…participants viewed these face-scene-face film sequences and were tasked with rating the valence of neutral faces.” Is partial as the task was to evaluate also emotional intensity/arousal.

INTRODUCTION

More on the Kuleshov effect story and its narration can be found in: Kolesnikov, A. (2020). The geocultural provenance of narratives: The case of the Kuleshov effect. Film History: An International Journal, 32(2), 55-79.

As for the alleged original experiment, please check whether when the contextual scene was that of a dead girl lying in a coffin the perceived emotion of the face was that of sadness.

Ref 15 is wrong, it refers to a study by P. Ekman while it should be related with the Kuleshov effect, please check and correct.

As for Barratt et al. study, participants were asked to evaluate the facial expressions in terms of valence and arousal, but also choosing among nine emotional labels, so in terms of categorization. Furthermore, authors state that “Barratt discovered that participant's judgments of facial expressions were indeed influenced by the emotional stimuli, thus affirming the existence of the Kuleshov effect.”; while this was true in terms of categorization, from a dimensional point of view this happened only for sadness and desire: neutral faces paired with sad contexts were rated as the most negative and least aroused, while neutral faces paired with desire contexts were perceived as the most positive and the most aroused.

Calbi et al., 2017 asked participants to evaluate the facial expression not only in terms of valence, but also of arousal and category. Please correct. And again, results clearly demonstrated the presence of a significant effect in terms of both valence and arousal but in the fear condition only. Moreover, participants tended to categorize the target person’s neutral facial expression choosing the emotion category congruent with the preceding context.

Regarding the passage on the different methodological approaches on neurocinematic research, I think that it should be better introduced as in its current form is somehow misleading. For instance, “In addition to reexamining the Kuleshov effect through behavioral experiments, cutting-edge neuroimaging techniques can enhance its validation by revealing its neural correlates.” Should be reframed in “In addition to reexamining the Kuleshov effect through behavioral experiments, THE USE OF cutting-edge neuroimaging techniques can enhance its validation by revealing its neural correlates.” Moreover, in the following sentence, which “current study” are the authors referring to?

Then, reading Kauttonen's introduction, I would say more that the authors describe the limitations of conventional methodology (e.g., GLM analysis) applied to fMRI studies with naturalistic stimuli (e.g., movies) in detecting brain activation related to annotated stimulus content when compared to model-free analysis methods than delineating an approach which uses naturalistic stimuli vs another which not.

As for Mobbs et al. fMRI study, authors should explain how the founded neural activations have been interpreted.

As for Calbi et al., 2019, behavioral results are partially reported as there were also valence and category evaluations.

METHODS

Regarding participants’ recruitment I have a curiosity, why such a long period to recruit them (from 2019 and 2023)?

How was the sample size determined (for both the experiments)?

- EXP 1

Regarding the participants, authors should report the age range (mean and standard deviation).

As for the validation of the emotional scenes, authors should report also results of post-hoc tests.

Did authors control for the “nature” of the content of emotional scenes across the three emotional conditions? For example, I see in Fig.1 that the happy emotional scene is depicted by a woman's face, are there similar stimuli (faces) in the other two conditions? Is there a control over the ratio between biological and non-biological ones?

Were the valence and emotional intensity scale VAS scales? How did participants answer?

The black screen at the end the experimental trial should be called Inter-trial interval (ITI) and not ISI (which is usually between two consecutive stimuli in the same trial).

Regarding the order of the conditions, why didn’t they follow a randomization procedure?

Did authors verify ANOVA assumptions?

How many participants did already know the Kuleshov effect?

Which analysis/statistical model was used to assess whether prior knowledge of the Kuleshov effect or viewing frequency did impact the observation of the Kuleshov effect?

- EXP 2

Why for the behavioral assessment in the fMRI experiment did authors use different scales (from -4 to + 4 instead of -1 +1 for valence and from 1 to 9 instead of 1-5 for arousal)? Why now authors employ the term arousal while in the first experiment they referred to emotional intensity? How did participants answer?

Why did authors this time use a repeated-measure ANOVA? Were assumptions verified?

ANOVA statistical results of Exp 1 and 2 should be accompanied by the effect size.

DISCUSSION

In general, I would suggest authors to rewrite the discussion to shorten it and make it smoother. The discussion opens, in fact, with a section (which we might call “general discussion”) that seems to exhaust all its relevant points in a superficial way only to find that it is followed by a series of in-depth sections. The authors may think not anticipating too much or stating that this is an initial general discussion to be followed by specific paragraphs devoted to each topic.

The authors might consider discussing why in Exp1 they found significant results only on the valence assessment, while the emotional intensity results were not significant. I think it is very interesting, especially because in experiment 2 they got significant differences in arousal as well. Regarding this, why authors state only that “the Kuleshov sequence triggered higher arousal in the fearful condition compared with the neutral condition” if this was true also for happy condition? I see that they dedicate a section on the difference between emotional intensity and arousal in another section, so maybe they can consider reframing or reorganizing the text to allow for a smoother reading (see my previous general point). Moreover, which question was exactly asked to participants to evaluate emotional intensity/arousal? How did authors explain the meaning of both to participants? (this should be added in methods section).

The section “Enhancing ecological validity of the Kuleshov effect with authentic films” does not add much relevant info but seems more like a repetition of many things explained already in the methods, can be shortened considerably to get straight to the point.

I do not entirely agree with this statement “In contrast to studies that employed neutral faces from the KDEF picture set without assessing distinct groups, our approach revealed no significant differences among faces in different conditions (F1,2 = 1.23, p = 0.305).” because in those cases stimuli were selected from an already validated dataset while this time authors built their own stimuli and they obviously had to validate them. Moreover, authors should check for the reported statistical values as they are different from those enlisted in the methods (page 13 line 277 of the revised manuscript with track changes).

Why did authors separate the discussion of neural correlats in two sections of the discussion?

MINOR:

line 268 (revised manuscript with track changes): and.

Line 502 (revised manuscript with track changes): not within but among

7. PLOS authors have the option to publish the peer review history of their article (what does this mean?). If published, this will include your full peer review and any attached files.

Reviewer #2: **Yes: **Jingyuan Ren

Reviewer #3: No

Reviewer #4: No

---

## [Author Response · Author response to Decision Letter 1]

19 Jun 2024

PONE-D-23-41138R1

Reexamining the Kuleshov effect: behavioral and neural evidence from authentic film experiments

PLOS ONE

Dear Dr. Wang,

Thank you for submitting your manuscript to PLOS ONE. After careful consideration, we feel that it has merit but does not fully meet PLOS ONE’s publication criteria as it currently stands. Therefore, we invite you to submit a revised version of the manuscript that addresses the points raised during the review process.

Editor comments: Fortunately, three reviewers agreed to comment on your manuscript. I ask you to revise the manuscript accordingly and to respond to each of the comments in a point-by-point manner.

We look forward to receiving your revised manuscript.

Kind regards,

Michael B. Steinborn, PhD

Section Editor

PLOS ONE

Response: We appreciate the positive comments from the editor. We have made revisions based on the Reviewers’ suggestions. 

Reviewers' comments:

Reviewer's Responses to Questions

Comments to the Author

1. If the authors have adequately addressed your comments raised in a previous round of review and you feel that this manuscript is now acceptable for publication, you may indicate that here to bypass the “Comments to the Author” section, enter your conflict of interest statement in the “Confidential to Editor” section, and submit your "Accept" recommendation.

Reviewer #2: (No Response)

Reviewer #3: (No Response)

Reviewer #4: (No Response)

2. Is the manuscript technically sound, and do the data support the conclusions?

Reviewer #2: Yes

Reviewer #3: Yes

Reviewer #4: Yes

3. Has the statistical analysis been performed appropriately and rigorously?

Reviewer #2: Yes

Reviewer #3: Yes

Reviewer #4: Yes

4. Have the authors made all data underlying the findings in their manuscript fully available?

Reviewer #2: Yes

Reviewer #3: Yes

Reviewer #4: Yes

5. Is the manuscript presented in an intelligible fashion and written in standard English?

Reviewer #2: Yes

Reviewer #3: Yes

Reviewer #4: Yes

6. Review Comments to the Author

Reviewer #2: Reexamining the Kuleshov Effect: Behavioral and Neural Evidence from Authentic Film Experiments

The current study addresses an exciting and significant topic, the Kuleshov effect, by integrating behavioral and neural evidence to explore the underlying neural mechanisms. The study's rigorous experimental design and logical framework are commendable. The results are noteworthy, potentially elucidating the behavioral and neural manifestations of the Kuleshov effect. However, several critical issues need to be addressed to enhance the study's impact.

Response: We appreciate the positive comments from the reviewer. We thank the helpful suggestions from the reviewer.

Major Issues:

1. Statistical Issues:

a. For whole-brain activation, the authors used FDR correction, but the cluster size threshold was set at above 5, which is quite small. Please provide justification for the chosen cluster size criterion.

We chose the default cluster size threshold in xjview because it allows us to observe a greater number of activated regions. This smaller cluster size is appropriate for our study as it helps to elucidate the underlying neural mechanisms of the Kuleshov effect. The relevant edits are listed as follows:

Methods, on page 22:

“This involved selecting positively activated brain regions, establishing the default cluster size threshold of five in xjview, which is deemed appropriate for our study to ensure that no critical activated regions are overlooked, necessary for understanding the neural mechanisms underlying the Kuleshov effect. Additionally, we applied false discovery rate (FDR) correction for multiple comparisons and set the significance level at p < 0.05."

b. The authors employed ANOVA followed by Fisher's Least Significant Difference for multiple comparisons. This method necessitates correction for multiple tests.

Response: We appreciate the reviewer's helpful suggestion. We applied Bonferroni correction in SPSS during the post hoc analysis. The relevant edits are listed as follows:

Methods, on page 14:

“Post hoc comparisons among the means were performed, employing a Bonferroni correction to account for multiple comparisons. “

Results, on page 24-25:

“Subsequent post hoc tests using Bonferroni correction confirmed significant differences in valence among each condition (p < 0.001), supporting the existence of the Kuleshov effect.

This correction revealed a significant main effect of emotional conditions on valence (F1.5 ,45.9 = 60.10, p < 0.001, η2 p = 0.667), with post hoc tests using Bonferroni correction revealing significant differences in valence among each condition (p < 0.001). These results reaffirmed the existence of the Kuleshov effect in an MRI scanner.

Post hoc tests using Bonferroni correction indicated significant arousal differences between fearful and neutral conditions (p < 0.001) and between happy and neutral conditions (p = 0.001). “

2. Experimental Design: On page 17, the fMRI experiment was conducted with a different presentation sequence compared to the behavioral study, which used randomization for the three conditions (happy, neutral, fearful). Could this sequential difference lead to interactions between trials? Given that emotions can persist over time, how can we ensure that the emotions induced by one trial do not influence subsequent trials?

Response: We apologize for the inadequate explanation of the randomization procedure. The trials in the happy condition were presented together, while the order of emotional conditions was randomized. Each participant experienced one of the six sequences: 'fearful-neutral-happy', 'fearful-happy-neutral', 'happy-neutral-fearful', 'happy-fearful-neutral', 'neutral-fearful-happy', or 'neutral-happy-fearful'. The frequency of each condition order was similar.

Since the ten trials in one emotional condition were presented together, emotions could persist over time before switching to another emotion. The inclusion of a jittered inter-trial interval (ITI) between trials also eliminated the possible interactions between trials. Relevant edits are listed as follows:

Methods, on page 19:

“Furthermore, we ensured a balanced order of emotional conditions. As there were three emotional conditions, each condition included ten trials. The ten trials for each emotional condition were presented together, while the order of emotional conditions was randomized across participants. Each participant experienced one of the six sequences: 'fearful-neutral-happy', 'fearful-happy-neutral', 'happy-neutral-fearful', 'happy-fearful-neutral', 'neutral-fearful-happy', or 'neutral-happy-fearful'. Additionally, the trials within each emotional condition were randomized. To mitigate the risk of emotional carryover effects from one trial to the next, we included a jittered ITI between each trial, allowing sufficient time for participants' emotional states to return to baseline. This approach helps ensure that the emotions induced by one trial do not influence subsequent trials."

3. The introduction and page 23 mention that the fMRI study shows that including jitters did not diminish the Kuleshov effect. Please omit this sentence as it detracts from the core objective of the study, which is to investigate the neural mechanisms of the Kuleshov effect.

Response: We appreciate the reviewer's insightful suggestion. We have omitted the sentence in the introduction and modified the sentences in the methods and results sections as follows:

Methods, on page 20:

“In the analysis of behavioral data, the primary objective was to examine the existence of the Kuleshov effect in the MRI scanner."

Results, on page 25:

“In this analysis, we sought to validate the existence of the Kuleshov effect in an MRI scanner."

4. In the fMRI analysis, three types of contrasts were used to investigate the neural basis of the Kuleshov effect. The second contrast, comparing happy or fearful conditions to the neutral condition, is similar to the third contrast, which compares Face 2 to Face 1 with the neutral condition. It would be more logical to either remove the second contrast or place it in the supplementary materials. The second contrast does not provide significant support for the current hypothesis and complicates the interpretation.

Response: We appreciate the reviewer's insightful suggestion. The second contrast has been placed in the supplementary materials. The relevant edits are listed as follows:

Supporting information, on pages 53-55:

“S2 Text. fMRI contrasting results. Comparison of brain activity for Face_2 between emotional and neutral conditions.

S7 Fig. Comparison of brain activity for Face_2 between emotional and neutral conditions. (A) Brain activity was obtained by subtracting Face_2 in the neutral condition from Face_2 in the fearful condition. (B) Brain activity was obtained by subtracting Face_2 in the neutral condition fromFace_2 in the happy condition. (p < 0.05, FDR-corrected, cluster size > 5 voxels).

S4 Table. fMRI Results: Face_2 in fearful or happy condition minus Face_2 in neutral condition. To probe the neural correlates of this Kuleshov effect bias, focus on how neutral faces exhibit distinct brain activation patterns when preceded by fearful or happy scenes. (p < 0.05, FDR-corrected, cluster size > 5 voxels). "

S2_Text:

“Supplementary results

Direct comparison of Face_2 between fearful or happy condition and neutral condition 

Examining the bars corresponding to the fearful and happy conditions in Fig 3C, it becomes evident that the Kuleshov effect introduces a noticeable context-dependent bias in emotional perception. This effect alters the interpretation of neutral faces based on accompanying emotional scenes. Our subsequent analysis aimed to probe the neural correlates of this Kuleshov effect bias, focusing on how neutral faces exhibit distinct brain activation patterns when preceded by fearful or happy scenes [1]. 

In the fearful condition compared to the neutral condition, the contrast between Face_2 revealed significant activation in 17 clusters, involving regions such as the bilateral cerebellum, fusiform gyrus (FG), parahippocampal gyrus (PHC), angular gyrus (AG), PCC, cuneus, precuneus, precentral gyrus, and postcentral gyrus (for additional AAL atlas labels, S4 Table and S7A Fig). These regions are well-established in facial and emotional processing, supporting that the Kuleshov effect reflects a contextual modulation of face perception.

Conversely, when contrasting Face_2 in the happy condition with those in the neutral condition, only one cluster of significant activation was detected, encompassing the right cuneus, right precuneus, and right calcarine fissure and surrounding cortex (CAL) (S4 Table and S7A Fig). These regions are linked to visual processing, suggesting that the influence of happy scenes on neutral faces is comparatively weaker than fearful scenes. “

5. Interpretation of Results: The author's interpretation of the whole-brain results seems somewhat far-fetched. My view is that the whole-brain results for the fearful condition, compared to the neutral condition, show activation in sensory and motion-related regions, possibly indicating the fight-or-flight response. In contrast, the happy condition, compared to the neutral condition, shows activation in the limbic system, likely related to the reward system.

Response: We appreciate the reviewer's insightful suggestion. We have revised the interpretation of fMRI results in the discussion. The relevant edits are listed as follows:

Discussion, on pages 36-39:

“Confirming the existence of the Kuleshov effect through fMRI

The Kuleshov effect encompasses two primary interpretations. The first is that viewers perceive a new meaning, such as emotion from a neutral face, as illustrated by the first bar in Fig 3A. The second interpretation is a context-dependent bias where, compared to a neutral condition, viewers perceive significant negative or positive emotions in emotional conditions, as demonstrated by the ANOVA results of Fig 3A. To confirm the existence of the Kuleshov effect through fMRI, these interpretations were examined through contrasting analyses.

In exploring the perceived new meaning of the second neutral face (Face_2), we investigated the neural correlates associated with this interpretation. The processing of the neutral face involved contrasting the activation in Face_2 with Face_1, revealing multiple regions of activation (Fig 4, S1-S3 Tables). Specifically, as a data exploratory study, our examination of the fearful condition revealed activations in various brain regions, including the insula, cuneus, ACC

---

## [Decision Letter · Decision Letter 2]

15 Jul 2024

PONE-D-23-41138R2Reexamining the Kuleshov effect: behavioral and neural evidence from authentic film experimentsPLOS ONE

Dear Dr. Wang,

Thank you for submitting your manuscript to PLOS ONE. After careful consideration, we feel that it has merit but does not fully meet PLOS ONE’s publication criteria as it currently stands. Therefore, we invite you to submit a revised version of the manuscript that addresses the points raised during the review process. **Editorial comment:** I am very excited about the excellent scientific discourse that reviewers and authors have engaged in here. Initially, I had difficulties finding reviewers, but we now have four excellent reviews where the authors received many valuable and constructive comments all aimed to improved the paper further. At present, three reviewers are satisfied and find the manuscript in proper shape, while R4 has some additional comments. To me, this seems we are now close to the final stage, and I ask you to revise the manuscript, providing a point-by-point response to the comments of R4. I have also some comments from my own reading (in my role as editor, and interested reader only) which the authors might consider. 

We look forward to receiving your revised manuscript.

Kind regards,

Michael B. Steinborn, PhD

Section Editor

PLOS ONE

Journal Requirements:

**Additional Editor Comments:**

First, I wish to clarify that my comments are intended to enhance the quality of the manuscript, not to criticise it. While I am not the foremost expert in your specific field, my view is aimed to offer the perspective of an interested reader who stands as a proxy for a broader audience, providing a scholarly perspective from slightly outside your domain. Having said that, this means - I do not expect you to adhere strictly to my remarks - however, I hope you may find certain points to be of use and value, which is exactly my intent.

**(-1-) introspective measures**

All reviewers, especially R1 and R4, have noted the issue of how to measure and what to measure to gain insight into the mechanisms underlying the Kuleshov effect. It appears to me that the question of whether objective and subjective measures represent different processes could be addressed more deeply in the discussion. Therefore, in line with the request of R4, I would also tend to suggest a more meticulous analysis of the components and characteristics of the processes reflected by both objective measures and introspective ratings in the discussion of the manuscript. To clarify, objective measures, such as physiological responses, provide quantifiable data that can complement subjective self-reports (see comments of R1). However, it is important to consider whether these measures converge to provide a complete picture and, if they do, how precisely they might do so. While physiological responses might indicate short-term general arousal (please correct me if I am wrong here), subjective reports may offer insights into the specific emotional quality experienced by participants, which can vary among individuals. Understanding the relationship between these measures is essential for a holistic interpretation of emotional responses. I suggest elaborating on this point more deeply.

**(-2-) valence vs. arousal**

The other point concerns the distinction between valence and arousal in subjective self-ratings. While valence refers to the positive or negative quality of an emotion, and arousal indicates the intensity or activation level of that emotion, it is not entirely clear how participants represent these dimensions introspectively when providing self-reports. Clear and consistent definitions and instructions for these terms are crucial for designing experiments and interpreting results. In line with the points raised by R1 and R4, I suggest addressing these issues more thoroughly in the discussion section. By explicitly addressing these methodological challenges, the study will not only improve the interpretability of its findings but also provide clear guidance and theoretical insights for future researchers. This deeper elaboration will certainly increase the manuscript impact.

**(-3-) the "semantics" of effect sizes**

R2 made an interesting point, suggesting reporting effect sizes for the ANOVA results on behavioural data, which implies the need for precise statistical reporting when using scales like VAS. I completely agree with this point. However, what R2 possibly wants to convey is the question of the semantic significance of scale differences. This is somewhat represented in effect sizes but not entirely. For example, if we find that the shoe sizes of German and French men are significantly different at 44.2 vs. 44.3, how meaningful is that difference? Even with a high effect size, should we simply say the shoe sizes are the same, or should we assert that they are different?

**(-4-) calibration of self-ratings**

R3 noted the issue of design and calibration differences in the experiments, asking why different scales were used for behavioural ratings (e.g., from -4 to +4 for valence and from 1 to 9 for arousal in one instance, but from -1 to +1 for valence and from 1 to 5 for arousal later on). I suggest elaborating on this point as a limitation in the discussion. Specifically, does this lead to interpretive problems, and if so, to what extent? What are the implications for future studies? R3 noted that the problem of calibration is even more complex, involving both rating calibration and individual differences in responses. This complexity includes scale anchoring effects and cultural differences, which affect rating tendencies and semantic differential effects when dimensions are interrelated (e.g., valence vs. arousal, energetic arousal vs. tense arousal, voluntary vs. involuntary mindwandering/inattention, to name examples of such dimensions, see Schumann et al. 2022, chap. 4). Cultural differences impact ratings in multiple ways: Western individuals might give overly positive responses, reducing discrimination in the lower range, while Eastern individuals may underutilise the upper end of the scale, and additionally, Western participants might approach multidimensional scales analytically, whereas Eastern participants might approach them holistically, whereby understanding these nuances is not a triviality but absolutely crucial for accurately interpreting data and designing future studies. I therefore warmly suggest elaborating on these issues (-1-) to (-4-) more deeply in the discussion. This may require additional effort for the final revision, but it will be worthwhile, as the end result will be a paper that not only presents empirical data but also addresses theoretical issues and proposes potential solutions. This will significantly enhance the impact of the study.

**(-5-) conclusion, contribution, future outlook**

At the final point in the manuscript, it may be appropriate to permit a bit of speculation, as scientific findings have two aspects to convey, reporting the hard-fact evidence and conveying a message to the world. Thus, the authors may, at this point, conjecture about how to arrange rating measures in future research to ensure accurate and meaningful measurements. From my layman perspective, this aspect of design seems crucial here and also R4 finally argued that the scales used are context-dependent and may only be valid in certain contexts, necessitating modifications. R4 suggested citing appropriate bibliographic references, particularly theoretical works, to support this point, especially those relevant to the measurement aspects of introspection, particularly regarding differences between ratings of emotional intensity and arousal, suggesting clarifying this in the discussion and including the relevant information in the methods section. In this regard, I would like to contribute a point. I suggest two theoretical works that, in my view, are highly relevant to address this point theoretically (though they may not seem directly related at first glance, but they definitely are when considering the larger picture). The first is a work by Schumann et al. (2022) (doi:10.3389/fpsyg.2022.867978, see chapters 4.2 to 4.5) and the second by Nisbett et al. (2001) (doi:10.1037/0033-295x.108.2.291), both theorising on verbal reports on mental processes, their relation to overt behavior and on measurement aspects of introspective ratings. I believe these works are capable of supporting the present claims and enhancing the discussions in the final section of the manuscript.

Reviewers' comments:

Reviewer's Responses to Questions

**Comments to the Author**

1. If the authors have adequately addressed your comments raised in a previous round of review and you feel that this manuscript is now acceptable for publication, you may indicate that here to bypass the “Comments to the Author” section, enter your conflict of interest statement in the “Confidential to Editor” section, and submit your "Accept" recommendation.

Reviewer #1: All comments have been addressed

Reviewer #2: All comments have been addressed

Reviewer #3: All comments have been addressed

Reviewer #4: (No Response)

2. Is the manuscript technically sound, and do the data support the conclusions?

Reviewer #1: Yes

Reviewer #2: Yes

Reviewer #3: Yes

Reviewer #4: Yes

3. Has the statistical analysis been performed appropriately and rigorously? 

Reviewer #1: Yes

Reviewer #2: Yes

Reviewer #3: Yes

Reviewer #4: Yes

4. Have the authors made all data underlying the findings in their manuscript fully available?

Reviewer #1: Yes

Reviewer #2: Yes

Reviewer #3: Yes

Reviewer #4: Yes

5. Is the manuscript presented in an intelligible fashion and written in standard English?

Reviewer #1: Yes

Reviewer #2: Yes

Reviewer #3: Yes

Reviewer #4: Yes

6. Review Comments to the Author

Reviewer #1: I want to thank the authors for constructively responding to the comments of the reviewers. The manuscript is stronger as a result and i believe that it will be widely cited. I intend to cite it in my own research.

Joe Magliano

Reviewer #2: All my concerns have been addressed satisfactorily in this revision. The authors have improved the clarity and quality of the manuscript. I appreciate the authors' efforts in responding to feedback. I am looking forward to seeing this paper.

Reviewer #3: (No Response)

Reviewer #4: I thank the authors for their effort in addressing my comments and suggestions. I think that the manuscript is now improved. In my opinion, some minor questions/clarifications remain:

- Please correct Page 5, lines 106-108: “This study further explored the contextual effects on emotion perception by asking participants to rate both the valence and arousal, and to explicitly categorize the emotion displayed by neutral faces following a Kuleshov sequence.”

- Please correct Page 8, lines 166-169:

EEG is mentioned for the first time, please write “electroencephalographic – EEG experiment”. Furthermore, the study is still not properly explained: “participants were tasked with rating the valence and arousal of neutral faces while simultaneously undergoing EEG recording, and subsequently to categorize the target person’s emotional state.” and it is not completely true that “the behavioral results replicated those of Calbi et al. (2017)” because this time there was a significant effect of Fear and Happy contexts on valence.

- Regarding sample size determination, please add its procedure in the main text.

- Please add mean age and SD for each group of participants (rating experiment, exp1 and exp2).

- Regarding the VAS scales, I see that the scale is from the Self-Assessment Manikin (SAM). Authors should add this info in the main text clearly stating if and how it was modified from the original one and citing the proper bibliographic reference.

- Regarding sub-groups comparison for prior knowledge of the Kuleshov experiment and film-watching frequency, subgroups have substantially different numerosity, I’m not sure they are statistically comparable.

- Page 32, lines 644-645: “participants were asked to rate the valence and the emotional intensity/arousal of the neutral face”

- As for the discussion, authors might consider to reorganize its first sections in the following way:

4.1) Enhancing ecological validity of the Kuleshov effect with authentic films (which is the novelty of your experimental contribution)

4.2) Confirming the existence of the Kuleshov effect through behavioral experiments

4.3) Dissociation between emotional intensity rating and physiological experience of arousal rating

4.4.) Inferring actors’ emotions under POV structure.

In this way the discussion of behavioral results, with a focus on differences between emotional intensity and arousal, is more fluid and not interrupted by “Enhancing ecological validity of the Kuleshov effect with authentic films” section.

7. PLOS authors have the option to publish the peer review history of their article (what does this mean?). If published, this will include your full peer review and any attached files.

Reviewer #1: **Yes: **Joseph Magliano

Reviewer #2: No

Reviewer #3: No

Reviewer #4: No

---

## [Author Response · Author response to Decision Letter 2]

18 Jul 2024

PONE-D-23-41138R2

Reexamining the Kuleshov effect: behavioral and neural evidence from authentic film experiments

PLOS ONE

Dear Dr. Wang,

Thank you for submitting your manuscript to PLOS ONE. After careful consideration, we feel that it has merit but does not fully meet PLOS ONE’s publication criteria as it currently stands. Therefore, we invite you to submit a revised version of the manuscript that addresses the points raised during the review process.

Editorial comment: I am very excited about the excellent scientific discourse that reviewers and authors have engaged in here. Initially, I had difficulties finding reviewers, but we now have four excellent reviews where the authors received many valuable and constructive comments all aimed to improved the paper further. At present, three reviewers are satisfied and find the manuscript in proper shape, while R4 has some additional comments. To me, this seems we are now close to the final stage, and I ask you to revise the manuscript, providing a point-by-point response to the comments of R4. I have also some comments from my own reading (in my role as editor, and interested reader only) which the authors might consider. 

We look forward to receiving your revised manuscript.

Kind regards,

Michael B. Steinborn, PhD

Section Editor

PLOS ONE

Journal Requirements:

Additional Editor Comments:

First, I wish to clarify that my comments are intended to enhance the quality of the manuscript, not to criticise it. While I am not the foremost expert in your specific field, my view is aimed to offer the perspective of an interested reader who stands as a proxy for a broader audience, providing a scholarly perspective from slightly outside your domain. Having said that, this means - I do not expect you to adhere strictly to my remarks - however, I hope you may find certain points to be of use and value, which is exactly my intent.

Response: We deeply appreciate the editor’s patience and efforts in overseeing the review process. Your guidance has significantly enhanced the quality of our manuscript. Regarding the editor's comments on the methodologies of subjective ratings and the relationship between brain and behavior, these issues have been thoroughly discussed in the limitations section to clarify their impact on our findings. Moreover, the overarching theme of the study, as highlighted by your comments—that of using both subjective and objective measurements to reexamine the existence of the Kuleshov effect—has been addressed more comprehensively in both the abstract and the conclusion sections. This ensures that readers can grasp the full extent of our research and its implications.

(-1-) introspective measures

All reviewers, especially R1 and R4, have noted the issue of how to measure and what to measure to gain insight into the mechanisms underlying the Kuleshov effect. It appears to me that the question of whether objective and subjective measures represent different processes could be addressed more deeply in the discussion. Therefore, in line with the request of R4, I would also tend to suggest a more meticulous analysis of the components and characteristics of the processes reflected by both objective measures and introspective ratings in the discussion of the manuscript. To clarify, objective measures, such as physiological responses, provide quantifiable data that can complement subjective self-reports (see comments of R1). However, it is important to consider whether these measures converge to provide a complete picture and, if they do, how precisely they might do so. While physiological responses might indicate short-term general arousal (please correct me if I am wrong here), subjective reports may offer insights into the specific emotional quality experienced by participants, which can vary among individuals. Understanding the relationship between these measures is essential for a holistic interpretation of emotional responses. I suggest elaborating on this point more deeply.

Response: We thank the editor for pointing out the significance of objective and subjective measures in our study. While our main aim was to reexamine the existence of the Kuleshov effect using authentic film materials, the intricate relationship between brain (objective measure) and behavior (subjective measure) warrants further exploration but was considered beyond the immediate scope of this study. However, it should be noted that the time-course of the generation of the Kuleshov effect presents an interesting question, where brain activation of Face_2 precedes the systematic level of valence rating. This sequential occurrence suggests a dynamic interaction between the neural activations and subsequent emotional assessments, aligning with the reviewers' interests in understanding how these two measurement types interrelate. The discussion of this brain-behavior relationship and the time-course of the generation of the Kuleshov effect are detailed in the limitations section. The relevant edits are outlined below:

Discussion, on pages 45-46:

“Lastly, in the timeline of a single trial in Experiment 2, the activation of Face_2 occurs earlier than the valence rating, indicating a disconnect between unconscious physiological activation and conscious emotional perception. Therefore, physiological responses represent immediate activation, while subjective reports reflect the participants' systematic assessment of emotional perception. This timing difference is crucial for interpreting the Kuleshov effect, as it suggests that physiological and subjective responses are part of a sequential process, rather than simultaneous phenomena. Despite these differences in the timing of measurements, both the localized brain activation patterns and the subsequent behavioral responses reveal the presence of the Kuleshov effect. Although understanding how the Kuleshov effect is generated—specifically, how brain responses transition to behavioral responses and result in complex emotional perception—is critical, a detailed exploration of this transition can further elucidate the underlying mechanisms of emotional processing in film viewing. It is beyond the scope of this study to fully explore this transition. Given the complexity of the brain-behavior relationship (Westlin et al., 2023), future research addressing this question may involve finite impulse response (FIR) time-course analysis within regions of interest (ROIs) or correlation analyses between ROIs and behavioral responses to help uncover the mechanisms behind the Kuleshov effect (Ren et al., 2023). This will significantly enhance our understanding of how different types of responses collectively contribute to the interpretation of cinematic effects. “

---------

In addition, as reminded by the editor, we think the validation of the existence of the Kuleshov effect with both subjective and objective measurements provides the complete picture for the current study. We have also revised the abstract and the conclusion to explicitly reflect the integration of both subjective and objective measurements. The relevant edits are outlined below:

Abstract, on pages 2-3:

“Overall, the study integrates film theory and cognitive neuroscience experiments, providing robust evidence supporting the existence of the Kuleshov effect through both subjective ratings and objective neuroimaging measurements. “

Conclusion, on page 46:

“We produced these film sequences and conducted two experiments. The results from the behavioral experiments indicate that viewers perceive emotions on neutral faces after watching the Kuleshov sequence, demonstrating that film editing can alter viewer perception by introducing new meanings and thus confirming the existence of the Kuleshov effect through subjective measurement. “

“The activation patterns in the insula, cuneus, precuneus, hippocampus, PCC, PHC, FG, and OFC not only support the existence of the Kuleshov effect from an objective measurement standpoint but also suggest that contextual framing is the mechanism underlying the Kuleshov effect. “

(-2-) valence vs. arousal

The other point concerns the distinction between valence and arousal in subjective self-ratings. While valence refers to the positive or negative quality of an emotion, and arousal indicates the intensity or activation level of that emotion, it is not entirely clear how participants represent these dimensions introspectively when providing self-reports. Clear and consistent definitions and instructions for these terms are crucial for designing experiments and interpreting results. In line with the points raised by R1 and R4, I suggest addressing these issues more thoroughly in the discussion section. By explicitly addressing these methodological challenges, the study will not only improve the interpretability of its findings but also provide clear guidance and theoretical insights for future researchers. This deeper elaboration will certainly increase the manuscript impact.

Response: We thank the editor for emphasizing the importance of clearly defining the terms valence and arousal in the methods section. To clarify, valence refers to the positive or negative quality of an emotion, while arousal indicates the intensity or activation level of that emotion. Another important aspect is how participants are instructed on these terms during the experiment. To address these concerns, we have added a detailed paragraph on the definitions of these terms in the Methods section and revised the experimental procedure to include specific details on how participants were educated about these concepts. Furthermore, we have included a new section on the limitations of subjective measurement tools in the Discussion section, addressing the methodological challenges raised by the reviewers. This addition not only enhances the clarity of the manuscript but also provides theoretical insights for future research. The relevant edits are outlined below:

Method, on page 16:

“2.1.2. Subjective Scales

The experiment employed the valence scale and emotional intensity scale to assess emotional perception. Valence refers to the positive or negative quality of an emotion as depicted in the actor’s performance, while emotional intensity measures the perceived degree of emotion in the actor’s performance. Both scales were Visual Analogue Scales, adapted from the Self-Assessment Manikin (SAM) (Bradley & Lang, 1994). To prepare the valence scale, the most negative, neutral, and positive expressions from the first line of the original SAM were selected and labeled from -1 to 1 beneath these pictures. Similarly, for the emotional intensity scale, the images from the second line of the original SAM were chosen, their order reversed, and they were labeled from 1 to 5 beneath these pictures. “

Method, on page 17:

“2.1.3. Procedure

Before the experiment, 59 participants were briefed on the procedure of the experimental task and the meanings of valence and emotional intensity. They then completed two practice trials to familiarize themselves with the procedure, using film sequences different from those in the formal experiment. If participants had questions about the procedure or the meanings of the scales, the instructions were repeated. “

Method, on page 19:

“2.2.1. Subjective Scales

The experiment employed the valence scale and arousal scale to assess emotional perception. Valence refers to the positive or negative quality of an emotion as depicted in the actor’s performance, while arousal measures the viewer's physiological response to the actor’s neutral face. Both scales were VAS, adapted from the SAM (Bradley & Lang, 1994). To prepare the valence scale, the images from the first line of the original SAM were chosen, their order was reversed, and they were labeled from -4 to 4 beneath these pictures. Similarly, for the arousal scale, the images from the second line of the original SAM were also chosen, their order was reversed, and they were labeled from 1 to 9 beneath these pictures. “

Method, on page 20:

“Before the experiment, 31 participants were briefed on the procedure of the experimental task and the meanings of valence and arousal. They then completed two practice trials to familiarize themselves with the procedure, using film sequences different from those in the formal experiment. If participants had questions about the procedure or the meanings of the scales, the instructions were repeated. “

Discussion, on pages 44-45:

“Thirdly, we must acknowledge the limitations of our tools for subjective emotion measurement. Emotions are complex and diverse, yet we have measured emotional perception from only three aspects: valence, emotional intensity, and arousal. Other dimensions such as categorization and dominance also exist, where overlaps between these emotional measurement tools occur (Calbi et al., 2017). Regarding the form of the tools, this study employed VAS to facilitate participants' understanding of emotions. Although VAS has been validated (Bradley & Lang, 1994), whether participants truly grasped the emotions through VAS remains to be explored. Concerning the range of the scales, this study used a smaller range in Experiment 1 and a larger range in Experiment 2. Although both experiments found similar statistical differences between emotional conditions with valence scales, as evidenced by the p-values and effect sizes, this indicated a robust observation of the Kuleshov effect across different ranges of scales. However, a smaller range can force choices for participants but may also overlook individual differences. In contrast, a larger range can capture individual responses and enhance the sensitivity of measurements by reducing the potential for ceiling or floor effects. Regarding participant characteristics, it should be noted that cultural backgr

---

## [Editor Report · Decision Letter 3]

22 Jul 2024

Reexamining the Kuleshov effect: behavioral and neural evidence from authentic film experiments

PONE-D-23-41138R3

Dear Dr. Wang,

We’re pleased to inform you that your manuscript has been judged scientifically suitable for publication and will be formally accepted for publication once it meets all outstanding technical requirements.

**Final editor comments.** The manuscript is now a truly excellent work. It contains significant contributions. It is an empirical study that builds on previous research and provides impulses for future research. Additionally, it is a study that deeply analyses the problem of measurement, and therefore I believe the present study will have a substantial impact. Although I am not an expert in this field, the interesting elaborations of the authors and all reviewers have made me somewhat of a semi-expert. I thank both the authors and the reviewers for this.

Kind regards,

Michael B. Steinborn, PhD

Section Editor

PLOS ONE
---

## [Editor Report · Acceptance letter]

26 Jul 2024

PONE-D-23-41138R3 

PLOS ONE

Dear Dr. Wang, 

I'm pleased to inform you that your manuscript has been deemed suitable for publication in PLOS ONE. Congratulations! Your manuscript is now being handed over to our production team.

Kind regards, 

on behalf of

Dr. Michael B. Steinborn 

Section Editor

PLOS ONE